# Assessing the ability to derive rates of polar middle-atmospheric descent using trace gas measurements from remote sensors

Niall J. Ryan[1], Douglas E. Kinnison[2], Rolando R. Garcia[2], Christoph G. Hoffmann[3], Mathias Palm[1], Uwe Raffalski[4], Justus Notholt[1]

[1] Institute of Environmental Physics, University of Bremen, Bremen, 28359, Germany
[2] Atmospheric Chemistry Observations and Modeling Laboratory, National Center for Atmospheric Research, Boulder, Colorado, USA
[3] Institute of Physics, Ernst-Moritz-Arndt-University of Greifswald, Felix-Hausdorff-Str. 6, 17489, Greifswald, Germany
[4] Swedish Institute of Space Physics, Box 812, SE-981 28 Kiruna, Sweden

*Correspondence to*: Niall J. Ryan (n_ryan@iup.physik.uni-bremen.de)

**Abstract.** We investigate the reliability of using trace gas measurements from remote sensing instruments to infer polar atmospheric descent rates during winter within 46 – 86 km altitude. Using output from the Specified Dynamics Whole Atmosphere Community Climate Model (SD-WACCM) between 2008 and 2014, tendencies of carbon monoxide (CO) volume mixing ratios (VMRs) are used to assess a common assumption of dominant vertical advection of tracers during polar winter. The results show that dynamical processes other than vertical advection are not negligible, meaning that the transport rates derived from trace gas measurements do not represent the mean descent of the atmosphere. The relative importance of vertical advection is lessened, and exceeded by other processes, during periods directly before and after a sudden stratospheric warming, mainly due to an increase in eddy transport. It was also found that CO chemistry cannot be ignored in the mesosphere due to the night-time layer of OH at approximately 80 km altitude. CO VMR profiles from the Kiruna Microwave Radiometer and the Microwave Limb Sounder were compared to SD-WACCM output, and show good agreement on daily and seasonal timescales. SD-WACCM CO profiles are combined with the CO tendencies to estimate errors involved in calculating the mean descent of the atmosphere from remote sensing measurements. The results indicate errors on the same scale as the calculated descent rates, and that the method is prone to a misinterpretation of the direction of air motion. The "true" rate of atmospheric descent is seen to be masked by processes, other than vertical advection, that affect CO. We suggest an alternative definition of the rate calculated using remote sensing measurements: not as the mean descent of the atmosphere, but as an effective rate of vertical transport for the trace gas under observation.

## 1 Introduction

The rate of the descent of air above the poles during winter has been an area of interest for some time because it influences the transport of trace gases from the thermosphere to the middle atmosphere (mesosphere and stratosphere) (e.g., Plumb et al., 2002; Engel et al., 2006; Smith et al., 2011; Manney et al., 2009). This vertical branch of the meridional circulation can

transport gases with high thermospheric volume mixing ratios (VMRs), and change the composition of the middle-atmosphere (e.g., Solomon et al., 1982; Allen et al., 2000; Hauchecorne et al., 2007; Smith et al., 2011; Garcia et al., 2014; Manney et al., 2009, 2015; Funke et al., 2014a, 2014b, 2017). The downward transport of nitrogen oxide (NO), produced around 110 km through energetic particle precipitation (EPP) events (e.g. Barth and Bailey, 2004; Randall et al., 2005, 2007,

2015), garners particular attention because $NO_x$ (NO + $NO_2$) catalytically destroys ozone in the stratosphere. Periods of strong atmospheric descent are known to coincide with the timing of sudden stratospheric warmings (SSWs) (e.g. Manney et al., 2003, 2009; Jackman et al., 2009, Siskind et al., 2010, Holt et al., 2013). Elevated VMRs of carbon monoxide (CO) and $NO_x$ have been observed to linger in the middle atmosphere when there is an exceptionally strong polar vortex after a strong SSW, helping to confine descending air at the pole (Randall et al., 2006). Randall et al. (2009) suggest that, in these cases,

VMRs of EPP-produced $NO_x$ are controlled more by mesosphere and lower thermosphere (MLT) descent rates than by the structure of the vortex. Siskind et al. (2016) found that due to downward transport of methane ($CH_4$) in years with strong, uninterrupted mesospheric descent, summertime upper stratospheric chlorine monoxide (ClO) is about 50 % greater than in years with strong horizontal transport.

A direct measurement of the mean vertical motion of air in the middle-atmosphere is currently not possible because of the

small velocities, and so an analysis of changes in measured tracer (relatively long-lived trace gas) VMRs offers a means to indirectly observe the vertical motion. This technique has often been used in combination with ozone measurements in the stratosphere to separate chemical and dynamical influences when determining ozone depletion (e.g. Proffit et al., 1990, 1993; Müller et al., 1996, 2003; Salawitch et al., 2002; Rösevall et al., 2007). Another indirect determination can be made using the diabatic circulation approach (e.g., Dunkerton, 1978; Solomon et al., 1986; Medvedev and Fomichev, 1994) but

this method is not discussed further here. The chemical lifetime of CO during polar night and the strong vertical gradient in its VMR make it a good tracer (Solomon et al., 1985; Allen et al., 1999; Lee et al., 2011). Water vapour ($H_2O$) is used to infer vertical motion (e.g., Straub et al., 2012) but a varying vertical gradient limits the altitudes at which it can be used (Lee et al., 2011). Nassar et al. (2005) used nitrous oxide ($N_2O$), $CH_4$, and $H_2O$ to infer rates of vertical motion in the upper stratosphere and lower mesosphere, and Bailey et al. (2014) used a combination of NO, $H_2O$, and $CH_4$ to derive profiles of

vertical motion in the middle atmosphere. A partial list of studies that have used tracers to calculate rates of vertical motion is given in Table 1, similar to Hoffmann (2012b) for studies using CO. The tracers used and the determined rates are also listed, along with the terminology used to describe the motion. The descent rates range from -100 m/day to -1200 m/day and show much variability, which can be expected as the studies were performed for different years and/or different times of the year, and at different locations (with the minus sign indicating descent rather than ascent). The altitude range over which the

rates were determined, and whether averaging was performed, is also shown in Table 1. It is important to note that an average over altitude can mask the higher descent rates that are found in the mesosphere. For example, Straub et al. (2012) show a descent rate of -325 m/day from averaged modelled wind profiles, between 0.6 hPa (~52 km) and 0.06 hPa (~68 km), whereas the individual wind profiles often show descent rates larger than -1000 m/day at 0.06 hPa.

A general assumption made in the derivations of the rates is that the observed change in a tracer VMR over time is caused by vertical advection. The assumption draws from the fact that the polar vortex edge acts as a barrier to lower latitude air and hinders horizontal mixing of air masses between the inside and outside of the vortex (Schoeberl et al., 1992, Manney et al., 1994), but limited mixing still occurs and defining the edges of the polar vortex is not straightforward (Manney et al., 1994, 1997; Harvey et al., 2009, 2015). During the formation/breakdown of the polar vortex at the beginning/end of winter, the transport barrier is weaker and the location of the vortex edge becomes much less well-defined (Manney et al., 1997), making it more difficult to identify inner-vortex air masses. This is also true for SSW events where the vortex is weakened, or breaks down and reforms, allowing increased mixing (Manney et al., 2009, 2015). When the vortex is well established, the edge definitions that rely on wind fields (such as scaled potential vorticity (e.g., Manney et al., 1994, 2007, 2011; Jin et al.,2006)) become less reliable in the mesosphere where wind observations are uncommon and reanalysis winds are known to be fallible (Manney et al., 2008a, b; Rienecker et al., 2011).

The aim of this study is to assess the limits of the above assumption when using tracer measurements from remote sounders to derive rates of vertical motion in the middle atmosphere. Measurements alone do not provide enough information to enable separation of the contributions to changes in tracer VMRs, and so an atmospheric model must be employed. The specified dynamics version of the Whole Atmosphere Community Climate Model (SD-WACCM) is used to determine the relative contributions to changes in CO VMRs during polar winter. The results are combined with daily average modelled CO to estimate the error associated with descent rates calculated assuming pure vertical advection of the tracer. Three commonly used representations of the data are assessed: a local area above a specific location (Kiruna, 67.8˚ N, 20.4˚ E, in this case), a zonal mean at a certain latitude (80˚ N is used as an example), and a polar mean (60˚ - 90˚ N). The winters of 2008/2009 and 2010/2011 are used in the study as an example of a winter with a strong SSW and a winter with a relatively stable vortex, respectively. The rate calculations were also performed using CO measurements from the Kiruna Microwave Radiometer (KIMRA) and the Microwave Limb Sounder (MLS) (not shown), and the results lead to the same conclusion. This was expected due to the level of agreement found in a comparison of the modelled and measured CO (Sect. 2.4).

Section 2 outlines the instruments and the model used in this study as well as their datasets. A brief comparison of the three CO datasets is provided to assess how well the model represents observations of the atmosphere. Section 3 investigates the trajectories of air parcels arriving in the Arctic during winter, and the evolution of the tendencies of CO VMRs in the Arctic middle atmosphere due to each component of the continuity equation. Section 4 shows the rates of vertical motion calculated using CO measurements and the above assumption, and estimates how these rates change when accounting for all parts of the continuity equation. Section 5 uses CO VMR tendencies from 2008 to 2014 to assess their relative importance during different months. Section 6 contains a discussion of the results and the limitations of the study, and suggests a different interpretation of the rates derived from tracer measurements. Section 7 provides concluding remarks.

## 2 Instruments, model, and data

### 2.1 KIMRA

KIMRA is a ground-based microwave remote sensor located at the Swedish Institute for Space Physics, Kiruna, Sweden (67.8° N, 20.4° E). The instrument is a passive remote sensor and radiances from atmospheric CO are measured at a frequency 230.54 GHz. Specific details on the instrument can be found in Raffalski et al. (2002) and Hoffmann et al. (2011). The current CO dataset from KIMRA covers Arctic winters from 2008 to 2015 with gaps corresponding to instrument non-operation and summer periods when CO VMRs in the middle atmosphere are too low to be accurately measured. The average altitude range of the data is 46 – 86 km and the vertical resolution is 15 – 18.5 km, depending on the altitude. Details on the measurement technique and CO inversion scheme are given in Ryan et al. (2017). The average precision (values can vary from one profile to another) of wintertime KIMRA CO VMRs range from 0.06 ppm at 46 km altitude to 2.7 ppm at 86 km. The average time resolution of a CO measurement is around 2 hours. KIMRA CO data presented in this work have been averaged to give daily profiles.

### 2.2 MLS

MLS is a microwave remote sensor aboard the Aura satellite, launched in July 2004, and is part of NASA's Earth Observing System. Atmospheric CO is retrieved from radiance measurement made in two bands of the 240 GHz radiometer. A description of the MLS instrument can be found in Waters et al. (2006) and details on the retrieval can be found in Pumphrey et al. (2007) and Livesey et al. (2008). The data used here are version 4.2 (Schwarz et al., 2015), which is described in Livesey et al. (2015). These CO profiles cover a pressure range of 215 – 0.0046 hPa (approximately 11 – 86 km) and have a maximum (largest) precision of 11 ppm at the highest (in altitude) pressure level. For the data used here, the vertical resolution is between 3.8 and 6.2 km, and the horizontal resolution is between 200 and 250 km. MLS data presented here are within ± 2° latitude and ± 10° longitude of Kiruna, and have been averaged to produce daily profiles.

### 2.3 SD-WACCM

The Community Earth System Model version 1 (CESM1), Whole Atmosphere Community Climate Model (WACCM), is a coupled chemistry climate model from the Earth's surface to the lower thermosphere (Marsh et al., 2013 and references therein). WACCM is a superset of the Community Atmosphere Model, version 4 (CAM4), and includes all of the physical parameterizations of CAM4 (Neale et al., 2013) and a finite volume dynamical core (Lin, 2004) for the tracer advection. The simulation of WACCM4 used in this study is run with specified dynamics (SD) fields, using meteorological analyses from the National Aeronautics and Space Administration (NASA) Global Modeling and Assimilation Office (GMAO) Modern-Era Retrospective Analysis for Research and Applications (MERRA) (Rienecker et al., 2011). The chemical component is based on version 3 of the Model for Ozone and Related Chemical Tracers (Kinnison et al, 2007). Garcia et al. (2014) discuss

an update to the absorption cross section for $O_2$, which increased modelled CO mixing ratios in the MLT, bringing them closer to observations.

SD-WACCM constrains the atmosphere towards observations below 50 km through nudging with data from the above-mentioned analysed meteorological fields. Above 60 km the atmosphere is fully interactive and between 50 and 60 km the nudging linearly decreases to zero. Details on the nudging of the temperature and wind fields in the model can be found in Lamarque et al. (2012) and references therein. The model also tends towards observations at altitudes above 60 km because the zonal-mean mesospheric winds and temperatures at higher altitudes have been shown to be strongly constrained by the stratosphere (Liu et al., 2010; McLandress, 2013), but precise agreement cannot be expected.

For the model runs used in this work, the pressure grid consists of 88 layers from the ground to the thermosphere (~133 km). The altitude resolution of the grid increases from ~0.1 km near the surface to ~3.5 km in thermosphere. The horizontal resolution is 1.9° x 2.5° in latitude and longitude. Model output of daily averages from 2008 to 2014 are used for this study.

**2.4 CO VMR comparison**

Figure 1 plots the CO VMRs from KIMRA, MLS, and SD-WACCM above Kiruna for 2008 through 2014. The aim of a comparison is to see whether the datasets capture the same variability in middle-atmospheric CO. The VMRs are plotted at five altitudes between 46 and 86 km. This is the average altitude range of KIMRA CO data and reaches the upper altitude limit of MLS CO data. The MLS and SD-WACCM VMRs are also plotted after the profiles have been smoothed with the KIMRA averaging kernels (Rodgers and Connor, 2003). This method allows a meaningful comparison of datasets that have significantly different vertical resolutions. A more comprehensive comparison of the current KIMRA and MLS data is given in Ryan et al. (2017). SD-WACCM data is here bilinearly interpolated to the location of Kiruna, but a significant change in the results isn't found when simply using the model coordinate grid point closest to Kiruna. The three CO datasets show agreement on seasonal and daily time scales. Agreement between the model and instruments on such time scales highlights the power of SD-WACCM as a tool for investigating trends as well as temporally short events in the atmosphere. A systematic difference is evident between MLS and SD-WACCM during the times of year when CO VMRs are rapidly increasing or decreasing, with SD-WACCM showing larger values of CO. The difference is most pronounced at higher altitudes and is predominantly during August/September and April/May. Table 2 lists the correlation and regression coefficients calculated for each data pairing. Regression coefficients are calculated accounting for errors in the abscissa and ordinate variables (York et al., 2004), with a 15 % error assumed for SD-WACCM CO VMRs. Correlations between KIMRA and smoothed MLS, and KIMRA and smoothed SD-WACCM, are $\geq 0.86$ at all altitudes, and MLS and SD-WACCM correlations are $\geq 0.88$. MLS and SD-WACCM were compared at other polar locations (not shown here) and display similar agreement. The values are similar to those found for earlier versions of the model and data (Hoffmann et al., 2012a), with differences mainly due to updates to the modelled CO (Garcia et al., 2014) and the data products (Livesey et al., 2015; Ryan et al., 2017).

## 3 Contributions to the CO continuity equation

### 3.1 The TEM continuity equation

The mass transport of the residual mean meridional circulation is well represented by the Transformed Eulerian Mean (TEM) formulation (Andrews and McIntyre, 1976), and is covered in detail in Andrews et al. (1987). The TEM circulation in the solstice seasons is dominated by flow from the summer to the winter pole, accompanied by downward and upward transport above the winter and summer poles, respectively. The TEM zonal-mean tracer continuity equation has been described in various works (e.g., Garcia and Solomon, 1983; Andrews et al., 1987; Brasseur and Solomon, 2005; Monier and Weare, 2005; Smith et al., 2011):

$$\frac{\partial \overline{\chi}}{\partial t} = -\overline{w*}\frac{\partial \overline{\chi}}{\partial z} - \overline{v*}\frac{\partial \overline{\chi}}{\partial y} + \bar{S} + \overline{X_{mol}} + \frac{1}{\rho_0}\nabla \cdot M + \overline{\frac{1}{\rho_0}\frac{\partial}{\partial z}\left(\rho_0 K_{zz}\frac{\partial \chi}{\partial z}\right)}, \tag{1}$$

where $\chi$ is the CO VMR, $\overline{w*}$ is the vertical component of the residual mean meridional circulation, and $\overline{v*}$, is the horizontal component. $\bar{S}$ is the zonal-mean net chemical production of CO and $\overline{X_{mol}}$ is the zonal-mean molecular diffusion of CO. $\nabla \cdot M$ is the divergence of the resolved eddy flux vector and describes the eddy transport of CO, with $\rho_0$ as the basic density. $K_{zz}$ is the diffusion coefficient due to unresolved small-scale gravity wave breaking, and the last component of Eq. (1) represents the transport of CO due to parameterised eddy flux divergence (from gravity waves). The right-hand-side (RHS) terms of Eq. (1) are calculated using daily averaged output from SD-WACCM and details on the exact equations can be found in Andrews et al. (1987). The value of $K_{zz}$ calculated with SD-WACCM depends, among other things, upon the Prandtl number (or more properly, the "turbulent Prandtl number"), which describes the ratio of momentum flux to heat flux. The Prandtl number is a property of the process whereby gravity waves dissipate when they "break" (see e.g., Fritts and Dunkerton, 1985, for more details). The Prandtl number is 2 for the model runs in this work (see Sect. 6) and is used in SD-WACCM to parameterise gravity wave breaking (Garcia et al., 2007). Finite differences in Eq. (1) are calculated as centred differences except at the boundaries of grids, where forward and backward differences are used.

The terms of Eq. (1) are renamed here as:

$$\frac{\partial \overline{\chi}}{\partial t} = adv\_w * + adv\_v * + chem + Xmol + Xedd + Xk_{zz}, \tag{2}$$

to simply express the tendencies of CO in the continuity equation. The change in zonal-mean CO VMR with time is a sum of the contributions from (following the RHS of Eq. (2)): vertical advection, horizontal advection, net chemical production, molecular diffusion, eddy transport, and unresolved eddy transport, which, for present purposes, is due to gravity waves.

### 3.2 Trajectories during Arctic winter

As a first step to examine the assumption of purely vertical tracer advection, the back-trajectories of air parcels during two Arctic winters are plotted in Figure 2. The parcels arrive at 5 altitudes between 46 and 86 km and at two locations, 67˚ N and 80˚ N. They are advected, from these locations, backwards in time over 60 days, in 4 hour steps, using $\overline{w*}$ and $\overline{v*}$ from SD-

WACCM. The starting date for the trajectory calculations (arrival date of the air parcels) is February 28[th], for 2009 and 2011. The winter of 2009 had a major SSW (during which the 10 hPa zonal circulation becomes easterly at 60° N) beginning on the 24[th] January, and 2011 had a relatively stable vortex throughout the winter. The results in Figure 2 are consistent with similar calculations in Smith et al. (2011) and Straub et al. (2012), for the general shape of the trajectory and in that the parcels do not originate above approximately 100 km. The parcels at 80° N arrive from higher altitudes due to a stronger vertical component of the circulation compared to 67° N. Conversely, the horizontal component of the circulation is stronger at 67° N and the parcels arriving there originate from lower latitudes compared to those arriving at 80° N. The magnitude of the TEM wind is larger for the higher altitudes, as is also shown in Smith et al. (2011), and the air parcels that arrive above 66 km altitude originate in the summer hemisphere. The parcels that arrive below this altitude, which could be considered as part of the Brewer Dobson circulation (Brewer, 1949), originate at latitudes closer to the equator. A clear reversal of the trajectory around the time of the SSW can be seen for the air parcel arriving at 66 km altitude, 80° N, because of temporary changes in the direction of $\overline{w*}$ and $\overline{v*}$. These changes cause the air parcel to reverse direction before starting to descend again. It is evident from Figure 2 that the circulations at the pole have varying degrees of vertical and horizontal components.

### 3.3 Tendencies of CO during Arctic winter

Figure 3, 4, and 5 plot the wintertime tendencies of CO (RHS of Eq. (2)) for 2008/2009 and 2010/2011, for the three scenarios of 67° N, 80° N, and a north polar average (60° – 90° N), respectively. The zonal mean tendencies are plotted as an 11-day running mean. Note that the labelled contours may be differently spaced in each panel, to match the range of values for individual tendencies. In the context of a point measurement at Kiruna, a full rotation of the vortex is on the order of 10 days (assuming a zonal wind speed of 20 m/s at 67° N). Relevant comments on the results are provided here but an in-depth analysis is not made as it is not the focus of the study. Molecular diffusion ($Xmol$) generally causes negligible changes in CO, compared to other process, below approximately 83 km, and shows little variation between different scenarios and winters. Above that, the magnitudes increase quickly, with tendencies < -0.1 ppm/day in the altitude range shown here. Unresolved eddy transport ($Xk_{zz}$) is also negligible below approximately 75 km, but can show tendencies < -0.2 ppm/day above that altitude for short times (order of a week). Significant variation is seen for the different winters. Both processes tend to cause a decrease in CO VMRs throughout the winter in the upper mesosphere, agreeing with results of Smith et al. (2011). For comparison, vertical advection ($adv\_w*$) at these altitudes shows positive tendencies ranging from < 0.2 to > 1.6 ppm/day. Changes in CO due to chemistry ($chem$) are small below approximately 70 km, but each scenario and year shows a sustained sink for CO during the winter in a layer at around 80 km altitude. The layer coincides with the location of a night-time layer of hydroxyl (OH) around 82 km altitude (Brinksma et al., 1998, Pickett et al., 2006, Damiani et al., 2010). OH is known as the dominant chemical sink for middle-atmospheric CO (Solomon et al, 1985). $chem$ tendencies are stronger at 80° N compared to 67° N, with magnitudes reaching more than 0.3 ppm/day in November and December 2010, ranging from approximately 10 % to 50 % of $adv\_w*$ over that time. The results suggest that CO chemistry cannot be ignored in the mesosphere during winter. Tendencies due to resolved eddy diffusion ($Xedd$) show the most variation

between positive and negative values, mainly at 67˚ N because of proximity to the edge of the polar vortex. The north polar average shows that $Xedd$ generally reduces CO VMRs during the winter and, above ~ 70 km, has magnitudes greater than 25 % of $adv\_w *$ for time scales of a week. The largest tendency in CO is from $adv\_w *$, and causes an almost constant increase in CO VMRs throughout the winter, before reversing when the TEM vertical wind changes direction in Spring

(visible in all $adv\_w *$ plots). The increase is due to the downward motion of air and the positive gradient of CO VMR with altitude. The tendency is stronger at 80˚ N compared to 67˚ N due to a stronger vertical component of the residual circulation at the higher latitude (Smith et al., 2011, and see Figure 2). A signature of the major SSW in 2009 can be seen in the $adv\_w *$ tendency for that year, with a decrease and eventual change to a negative tendency. A negative tendency generally indicates ascent of air at this time. For some time directly afterwards, the tendency has a stronger positive magnitude than

before. This agrees with observations of stronger vertical motion above the pole after a SSW (see references in Table 1). There is also a brief change to a negative $adv\_w *$ at 80 N, around 80 km altitude, in early January 2011. This coincides with a relatively strong positive value for $Xedd$ at the same time and location, indicating strong wave activity. The CO tendency from horizontal advection ($adv\_v *$) is negative almost everywhere. This is expected, considering the direction of $\overline{v *}$, toward the winter pole, and the low-to-high gradient of CO from lower to higher latitudes in the winter hemisphere. The

magnitude of the tendency decreases in spring in each scenario and year. but a change of sign is not obvious by the end of April. The advection tendencies show maximum values around $70 - 80$ km for two main reasons. The first is the larger magnitude of the TEM circulation, compared to lower altitudes, before there is a turnaround in the direction of the circulation at higher altitudes, at which point the circulation changes from poleward and downward to poleward and upward (e.g., Lieberman et al., 2000; Smith et al., 2011). The turnaround point is at approximately 95 km in WACCM (Smith et al.,

2011). The second is the generally increasing vertical gradient of CO with altitude (see Eq. 1). At 67˚ N, the magnitudes of $adv\_v *$ are roughly half that of $adv\_w *$, and at 80˚ N they are roughly one fifth. Considering this observation alone, changes in CO VMRs cannot be attributed solely to vertical advection.

## 4 Rates of vertical motion with SD-WACCM CO

In this section, the rates of vertical motion are calculated, using CO profiles from daily averaged SD-WACCM output, by

two methods. The first method assumes that observed changes in CO VMRs are due to vertical advection alone. This is a commonly used method (see Table 1) and involves tracking the altitude of a chosen VMR of CO over time and then performing a linear regression on the data of altitude ($z$) vs. time ($t$). The rate for a given date is calculated here by performing a regression on the CO data within ±5 days of the date. This gives an 11-day running mean of the rate of vertical motion. The calculation can be done for multiple VMRs, to retrieve a vertical profile of the rates of vertical motion, as in

Bailey et al. (2014). The term used here to denote the rate of vertical motion calculated using CO VMR is $w_{CO}$, following Hoffmann (2012b). The second method used here includes information on changes in CO VMRs due to all terms of the continuity equation in Sect. 2. Before tracking a CO VMR from one time step to the next, the VMR is adjusted using the

tendencies of the continuity equation, except for $adv\_w *$, for the corresponding time frame. This accounts for changes in CO VMR from other processes as one tracks its movement due to vertical advection. The resulting rate is called $w_{CO}\ corrected$. This could be considered a crude approach, combining daily averaged CO output with CO tendencies calculated using the TEM formalism, but the aim here is to provide an estimate of the errors that may be incurred by

neglecting influences on CO other than vertical advection. In any case, the results involving $w_{CO}\ corrected$ are discussed in a qualitative manner, instead of for quantitative error analysis. To illustrate the difference between $w_{CO}$ and $w_{CO}\ corrected$, example algebraic expressions for the rates between a time step $n$ and $n+1$ are given: Eq. (3), (4), and (5). In practice, $w_{CO}$ and $w_{CO}\ corrected$ were calculated by performing a linear regression on the 11-day altitude vs. time data, including errors in the estimated altitude of the measured CO VMRs. Equation (3), (4), and (5) refer to scalar (values at a given

altitude/latitude/longitude) VMRs and tendencies, in contrast to Eq. (1) and (2), which refer to three-dimensional variables.

$$w_{CO}(z,t) = \frac{z_{n+1}-z_n}{t_{n+1}-t_n}\bigg|_{\bar{\chi}} \tag{3}$$

$$w_{CO}\ corrected(z,t) = \frac{z(\bar{\chi}_{n+1})-z(\bar{\chi}_n)}{t_{n+1}-t_n} \tag{4}$$

where,

$$\bar{\chi}_{n+1} = \bar{\chi}_n + \left(\frac{1}{2}\sum_{i=n}^{n+1}[adv\_v *+chem+Xmol+Xedd+Xk_{zz}]\right)(t_{n+1}-t_n) \tag{5}$$

The results for winters 2008/2009 and 2010/2011 are shown. For each winter, $w_{CO}$ and $w_{CO}\ corrected$ are calculated as: a local value above Kiruna, a zonal mean at 80° N, and a polar mean (60° – 90° N), and plotted in Figure 6, 7, and 8, respectively. The methods above were found to be unreliable when there are very low CO VMRs, and gave unrealistic rates of motion. This was more likely to occur at lower altitudes where CO VMRs are relatively low. Sometimes a CO VMR could not be followed within the specified altitude range (no extrapolation was used) and so there are a few locations with no

rate information. The differences between the two rates, $w_{CO}\ corrected - w_{CO}$, are also shown in each figure. $w_{CO}$ was also calculated using the data from KIMRA, for 67° N, and from MLS, for the zonal mean at 80° N and the north polar mean. The results are included in Figures 6 – 8 for comparison. The $w_{CO}$ values from SD-WACCM and from the instruments show good agreement, with some differences that would be expected due to the levels of agreement of the CO VMRs (see Sect. 2). Considering the calculations using SD-WACCM, there are three main qualitative points, common to each scenario and year,

that are evident from the results. The same conclusions are reached when using KIMRA and MLS in place of SD-WACCM. First, the values of $w_{CO}$ are generally of a smaller magnitude than $w_{CO}\ corrected$ during winter, meaning the calculated rates of descent are stronger if one accounts for CO tendencies other than vertical advection. This makes sense because, as seen in Figures 3-5, the other transport terms of the continuity equation (and the chemical loss term) tend to oppose the vertical advection term. In other words, the results indicate that the "true" rate of atmospheric descent is masked by sinks of

CO, and by transport processes that oppose the tendency due to vertical advection. Second, the differences between the two rates are often of the same order as $w_{CO}$. Third, the signs of $w_{CO}$ and $w_{CO}\ corrected$ are often opposite, meaning the calculated direction of air motion is prone to change when accounting for CO tendencies other than vertical advection. In

each example for 2008/2009, the magnitude of the positive (upward) motion around the time of SSW is decreased for $w_{CO}$ corrected compared to $w_{CO}$. After the SSW, and into March, the strongest descent values are seen around 70 – 80 km in $w_{CO}$ corrected, compared to values of ascent seen in $w_{CO}$ at the same location.

## 5 Relative strengths of CO tendencies, by month

To give an idea of the relative influence each tendency has on middle-atmospheric CO VMRs each month, daily CO tendencies from 2008 to 2014 are used to provide relative values of their monthly mean. For a given tendency, the daily values are separated by calendar month and averaged, to give a monthly mean tendency. The daily sums of the absolute values of all tendencies are also separated by month and averaged, to give a monthly mean total absolute tendency. The monthly mean tendencies are then normalised by the monthly mean total absolute tendency, and will be referred to here as

relative strengths. Using absolute values for normalisation retains the sign of the individual tendencies and avoids a large spread in the results when there is a small denominator (i.e., when the tendencies cancel each other and their sum is near zero). The relative strength of the sum of all tendencies, excluding $adv\_w *$, is also included as a variable. This variable will be referred to here as "other processes" and can be directly compared to the relative strength of $adv\_w *$, to judge the influence of vertical advection compared to all other processes. The results are shown for the north polar average (60˚ –

90˚ N) in Fig. 9, and the south polar average (60˚ – 90˚ S) in Fig. 10. For the north, tendencies corresponding to 10 days directly before and after a SSW (starting on January 22[nd], 24[th], and 26[th], 2008, 2009, 2010, respectively, and February 6[th], 2013) are excluded from the calculation, as well as the SSW start-date. Ten days was chosen to remove effects directly before and after a SSW, but signatures of the events remain in the data. The information from before and after an SSW is used to separately calculate the relative strengths for these times and is shown in Fig. 11.

For the north polar average in Fig. 9, the relative strength of other processes is everywhere negative from October to March and the relative strength of $adv\_w *$ is positive. Both $adv\_w *$ and other processes show a change of sign in April. The relative strength of $adv\_w *$ reaches a maximum value of 0.8 at lower altitudes in October and November, and decreases with altitude to approximately 0.5 at 86 km for these months. The relative strength of other processes shows an opposite trend: negative values strengthening with altitude to approximately -0.4. At the lower limit of the altitude range in January,

$adv\_w *$ shows a lower relative strength than other processes due to a strong $Xedd$ influence. This is likely the "left-over" influence of the SSWs in the data. By March, the relative strength of chem is prominent below 65 km, and the magnitude of $adv\_w *$ is matched by other processes. By April, the residual mean circulation has reversed direction and $adv\_w *$ has changed sign at most altitudes. The negative value of chem is then dominant at lower altitudes, and there is a stronger positive tendency above ~80 km (from photolysis of carbon dioxide ($CO_2$)).

There are no months where the relative strength of other processes can be considered negligible compared to the relative strength of $adv\_w *$. The closest approximations of this situation are at 50 km altitude in October and at 46 km altitude in November, when other processes contributes 13.7 % and 9.6 % of $adv\_w *$, respectively. These percentages then vary

significantly with altitude. For October, the value increases to 18.6 % at 46 km, 22.5 % at 60 km, and is 61.13 % at 80 km. For November, the value increases to 34.4 % at 54 km, and is 70.8 % at 80 km.

The results for the south polar average, in Fig. 10, are qualitatively similar to those for the north. The relative strength of $adv\_w *$ shows a maximum of ~0.8. Both hemispheres show a peak in *chem* at 80 km for most of winter (see Sect. 3.3). The

relative strength of $Xedd$ is not as prominent at the south as the north, likely due to the higher stability of the southern polar vortex. The points at which the relative strength of *other processes* is smallest compared to $adv\_w *$ are at 56 km in April (8.3 %) and at 46 km in May (6.8 %). For April, the value increases to 22.5 % at 46 km and 21.5 % at 66 km, and is 56.9 % at 80 km. In May, the value increases to 16 % at 54 km, and is 69.1 % at 80 km.

For the 10 days directly before and after SSWs, in Fig. 11, the relative strength of $adv\_w *$ is less than 0.5 at all altitudes.

$Xedd$ is strong below 60 km, such that the relative strength of *other processes* has a larger magnitude than that of $adv\_w *$ at many altitudes. The relative strength of $adv\_w *$ shows a more oscillatory structure with altitude, and there is a local minimum at about 70 km in the data for 10 days after SSWs. There is also a positive peak in the relative strength of $Xk_{zz}$ after SSWs at this altitude.

Aside from considering what value would classify as negligible, the significant variation in strength of *other processes*

compared to $adv\_w *$, over altitude, adds complexity to the method of following a tracer over an altitude range to determine the descent rate. One must also consider that while this section discusses monthly averaged data, tracers are often followed for several days to determine the changes in altitude over that time, and that the magnitudes of each tendency can vary significantly over this time scale (see Fig. 3, 4, and 5).

## 6 Discussion

The results of the previous sections, using SD-WACCM, are clear on one indication, that the assumption of observed changes in CO VMRs being solely due to vertical advection is not a valid one. What is not obvious, is what the rates inferred from the behaviour of tracer isolines ($w_\chi$, under the nomenclature used here) represent. When making observations on timescales of weeks, the TEM offers a suitable representation of the governing dynamics. The vertical velocities calculated by observing CO, however, are smaller than the magnitudes of the TEM vertical wind, $\overline{w *}$, found with SD-WACCM.

Figure 12 shows the polar average ($60° – 90°$ N) $\overline{w *}$ for the winters of 2008/2009 and 2010/2011, calculated using daily averaged SD-WACCM output. Comparing this to the values of $w_{CO}$ and $w_{CO}$ *corrected* in Fig. 8, it is clear that the rates derived from CO values are generally of lower magnitude than $\overline{w *}$. This agrees with the results of Hoffmann (2012b). $w_{CO}$ *corrected* more closely matches the sign and general pattern of $\overline{w *}$, but does not reach the semi-persistent descent rates of -1.2 km/day between 60 and 70 km, or the magnitude of the enhanced rate of descent (< -1.6 km/day) after the SSW

in 2009. Similar results are observed when using KIMRA and MLS CO profiles to derive vertical velocity, instead of SD-WACCM output (not shown here).

Some of the difference may be attributed to the time resolution, or assumed parameters, of the SD-WACCM output used in this work. Meraner and Schmidt (2016) showed that the tendency of NO due to $\overline{w*}$ can be quite different (by > 1 ppm/day at 90 km) when calculated using 6-hourly averaged output or daily averaged output from HAMMONIA (Hamburg Model of the Neutral and Ionized Atmosphere). Meraner et al. (2016) showed that $NO_x$ transport in the mesosphere is highly sensitive

to the strength of the gravity wave source. The amplitudes of gravity waves influence the altitude at which the waves break and deposit their momentum, which in turn affects the vertical profile of $\overline{w*}$. Garcia et al. (2014) found that using a Prandtl number of 4 for WACCM (see Sect. 3.1), instead of 2 (as used here), gave better agreement with polar CO profiles from satellite instruments (the Atmospheric Chemistry Experiment Fourier Transform Spectrometer (ACE-FTS) on SCISAT-1, and the Michelson Interferometer for Passive Atmospheric Sounding (MIPAS) on Envisat). The opposite was found for

tropical and mid-latitude CO profiles, and Garcia et al. (2014) consider that deficiencies in the modelling of diffusive eddy transport may be due to problems in producing the correct latitudinal profile of $K_{zz}$, which cannot be addressed by simply adjusting the Prandtl number.

As the results here using SD-WACCM indicate that the commonly used approximation of $\overline{w*}$ with $w_\chi$ (using tracer observations) is not valid, we suggest an alternative interpretation of $w_\chi$: as an effective rate of vertical transport for the trace

gas $\chi$. The interpretation still concerns the descent of a tracer through the middle atmosphere, but allows for a chemical sink, or for processes other than vertical advection to influence the descent rate, e.g., horizontal advection or an increase in unresolved eddy transport after a SSW. Hoffmann (2012b) put forward a similar interpretation for $w_{CO}$, but with an assumption that the overall dynamic effects on CO are representative for mesospheric air, and so $w_{CO}$ is representative of $w_\chi$ for all tracers. The results here do not confirm that assumption, as it would require tracers to have the same horizontal and

vertical VMR gradients (see Sect. 2.5 and references therein). The consistent negative chemical tendency of CO, seen here in a layer around approximately 80 km, also indicates the need to account for the behaviour of chemical sinks, even during polar night.

## 7. Conclusion

The aim of this work was to assess how well polar atmospheric descent rates can be derived from remote sensing

measurements of atmospheric trace gases. Tendencies of middle atmospheric CO were calculated using output from SD-WACCM for the years 2008 to 2014, within 46 – 86 km altitude, and used to evaluate the relative influence of processes involved in the TEM CO continuity equation. The results show that dynamical processes other than vertical advection cause non-negligible changes in CO VMRs during winter, and particularly directly before and after sudden stratospheric warmings when eddy transport can become dominant. Significant changes in CO tendencies from SD-WACCM occur on the order of

days. The results also show a chemical sink for CO, present throughout polar night, due to the layer of night-time OH at approximately 80 km. Modelled CO profiles were used in combination with the tendency data to provide a qualitative estimate of the errors that can be incurred when assuming pure vertical advection of CO. Rates of atmospheric motion were

calculated when assuming only vertical advection, and corrected rates were calculated by including tendency information for all processes. The differences between the two results are of the same order as the calculated rates, and the rates are prone to showing opposite directions for the mean vertical wind. The corrected rates more closely match the TEM vertical wind velocity from SD-WACCM, but both results using CO show smaller magnitudes relative to the TEM vertical wind, in agreement with the work of Hofmann (2012b). The "true" rate of atmospheric descent appears to be masked by sinks of CO, and by transport processes that oppose the tendency due to vertical advection. Monthly mean relative tendencies for CO show that the summed magnitude of processes other than vertical advection can constitute a large fraction of the changes in CO VMR. For a given month, the summed magnitude of the other processes, relative to vertical advection, changes by several tens of percent over the altitude range under investigation. The results here suggest that there are no months during polar winter when vertical mean advection dominates the budget of CO to such an extent that vertical mean velocity can be accurately derived within the altitude range.

An assessment using SD-WACCM indicates that a commonly used approximation of the vertical mean velocity of the atmosphere, $\overline{w}*$, using tracer (CO in this case) isolines is not valid, and an alternative interpretation of the rates derived from trace gas measurements is suggested: an effective rate of vertical transport for the given trace gas. Such an interpretation still concerns the descent of trace gases from the mesosphere and thermosphere, but allows for chemical sinks, and changes in VMR from dynamical processes other than vertical advection. Due to possible differences in the behaviour of chemical sinks and VMR gradients, it is not clear whether the rate of vertical transport for one tracer is representative of the rate for another. Continuous ground-based and satellite measurements of trace gases remain an essential tool in understanding the short- and long-term evolution of the middle atmosphere, as well as for the validation and parameterisation of atmospheric models.

**Author contributions**

Mathias Palm and Christoph G. Hoffmann outlined the project. Niall J. Ryan designed the method. Douglas E. Kinnison performed the SD-WACCM runs and provided the output. Rolando R. Garcia provided scripts to calculate TEM variables from WACCM output. Uwe Raffalski maintains and operates KIMRA, and provided the instrument spectra. Niall J. Ryan performed the study with valuable insight and interpretation of results provided by Douglas E. Kinnison and Rolando R. Garcia. Justus Notholt oversaw project development. Niall J. Ryan prepared the paper with contributions from co-authors.

**Competing interests**

The authors declare that they have no conflict of interest.

**Acknowledgements**

A 3-month research stay by Niall J. Ryan at the National Centre for Atmospheric Research, Boulder, Colorado, was funded by the University of Bremen, under the German Excellence Initiative (ABPZuK-03/2014). Instrument data used in the project was developed and acquired through funding from the German Federal Ministry of Education and Research (BMBF) through the research project: Role Of the Middle atmosphere in Climate (ROMIC), sub-project ROMICCO, project number 01LG1213A. The manuscript creation and editing was supported through ROMICCO. We would like to thank the National Center for Atmospheric Research (NCAR) for being so accommodating. The National Center for Atmospheric Research is sponsored by the US National Science Foundation. We express our gratitude to the MLS team for making their data available. A special thank you to Sophia, for all the smiles.

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

| reference | tracer | location | instrument | rate of vertical motion [m/day] | approximate altitude(s) | time of year | terminology |
|---|---|---|---|---|---|---|---|
| Allen at al. (2000) | CO | Antarctica | satellite | -250 at 60° S. -330 at 80° S | 30 - 50 km average | April/May average, 1992 | atmospheric descent |
| Forkman et al. (2005) | CO | NH mid-latitudes | ground-based | +250 to +450 | 60 - 95 km | Spring, 2002 | mesospheric circulation |
| Forkman et al. (2005) | CO | NH mid-latitudes | ground-based | 0 to -300 | 60 - 95 km | Autumn, 2002 | mesospheric circulation |
| Nassar et al. (2005) | $CH_4$, $H_2O$ | Arctic | satellite | -150 | upper straosphere | Feb/Mar, 2004 | atmospheric descent |
| Nassar et al. (2005) | $CH_4$, $H_2O$ | Arctic | satellite | -175 | lower mesosphere | Oct 2003 to Feb 2004 average | atmospheric descent |
| Hauchercorne et al. (2007) | $NO_2$ | Arctic | satellite | from -600 in Jan, to -200 in Mar | 45 - 70 km | 20 Jan to 10 Mar, 2004 | descent of NO2 layer |
| Funke et al. (2009) | CO | Arctic | satellite | -350 to 400 | 50 - 70 km | Sep/Oct, 2003 | polar descent |
| Funke et al. (2009) | CO | Arctic | satellite | -200 to -300 | 40 - 70 km | Nov/Dec, 2003 | polar descent |
| Funke et al. (2009) | CO | Arctic | satellite | -1200 | mesosphere | after SSW, 2004 | polar descent |
| Di Bagio et al. (2010) | CO | Arctic | ground-based | -200 to -300 | descent from 58 - 62 km | after SSW, 2009 | descent of air |
| Di Bagio et al. (2010), from Orsolini et al. (2010) | $H_2O$ | Arctic | ground-based | -200 to -300 | descent from 59 and 62 km | after SSW, 2009 | descent of air |
| Straub et al. (2012) | $H_2O$ | Arctic | ground-based | -350 | 52 - 68 km average | 5 Feb to 5 Mar average, 2010 | polar descent |
| Straub et al. (2012) | $H_2O$ | Arctic | satellite | -360 | 52 - 68 km average | Jan/Feb/Mar average, 2010 | polar descent |
| Bailey et al. (2014) | NO, $H_2O$, $CH_4$ | Arctic | satellite | up to -1000 | 40 - 90 km | after SSW, 2013 | atmospheric descent |

**Table 1: A partial list of studies that used atmospheric tracers to derive rates of vertical air motion in the atmosphere.**

|  | KIMRA-MLS | KIMRA-MLSsmooth | KIMRA-WACCM | KIMRA-WACCMsmooth | MLS-WACCM | MLSsmooth-WACCMsmooth |
|---|---|---|---|---|---|---|
| **86km** | 0.87, 0.95 ± 0.03 | 0.93, 1.26 ± 0.05 | 0.81, 1.05 ± 0.01 | 0.86, 1.25 ± 0.01 | 0.90, 1.00 ± 0.02 | 0.89, 0.93 ± 0.05 |
| **76km** | 0.90, 1.46 ± 0.03 | 0.94, 1.27 ± 0.03 | 0.89, 1.18 ± 0.01 | 0.89, 1.26 ± 0.01 | 0.91, 0.78 ± 0.01 | 0.87, 0.78 ± 0.02 |
| **66km** | 0.90, 1.27 ± 0.02 | 0.95, 1.36 ± 0.03 | 0.85, 1.20 ± 0.01 | 0.90, 1.26 ± 0.01 | 0.90, 0.83 ± 0.01 | 0.85, 0.72 ± 0.01 |
| **56km** | 0.91, 1.01 ± 0.02 | 0.95, 1.20 ± 0.02 | 0.88, 0.95 ± 0.01 | 0.90, 01.00 ± 0.01 | 0.89, 0.86 ± 0.01 | 0.86, 0.70 ± 0.01 |
| **46km** | 0.88, 0.38 ± 0.01 | 0.91, 1.23 ± 0.04 | 0.89, 0.30 ± 0.01 | 0.90, 0.95 ± 0.02 | 0.88, 0.86 ± 0.01 | 0.83, 0.63 ± 0.01 |

**Table 2: "correlation coefficient, regression coefficient ± error" for comparisons of daily CO VMRs from KIMRA, MLS, and SD-WACCM above Kiruna for 2008 through 2014. The abscissa variable is the first-named of each instrument pairing. A regression coefficient > 1 (< 1) indicates a larger (smaller) range in the abscissa variable. Figure 1 shows the time-series' of each CO VMR dataset. See Sect. 2.4 for details.**

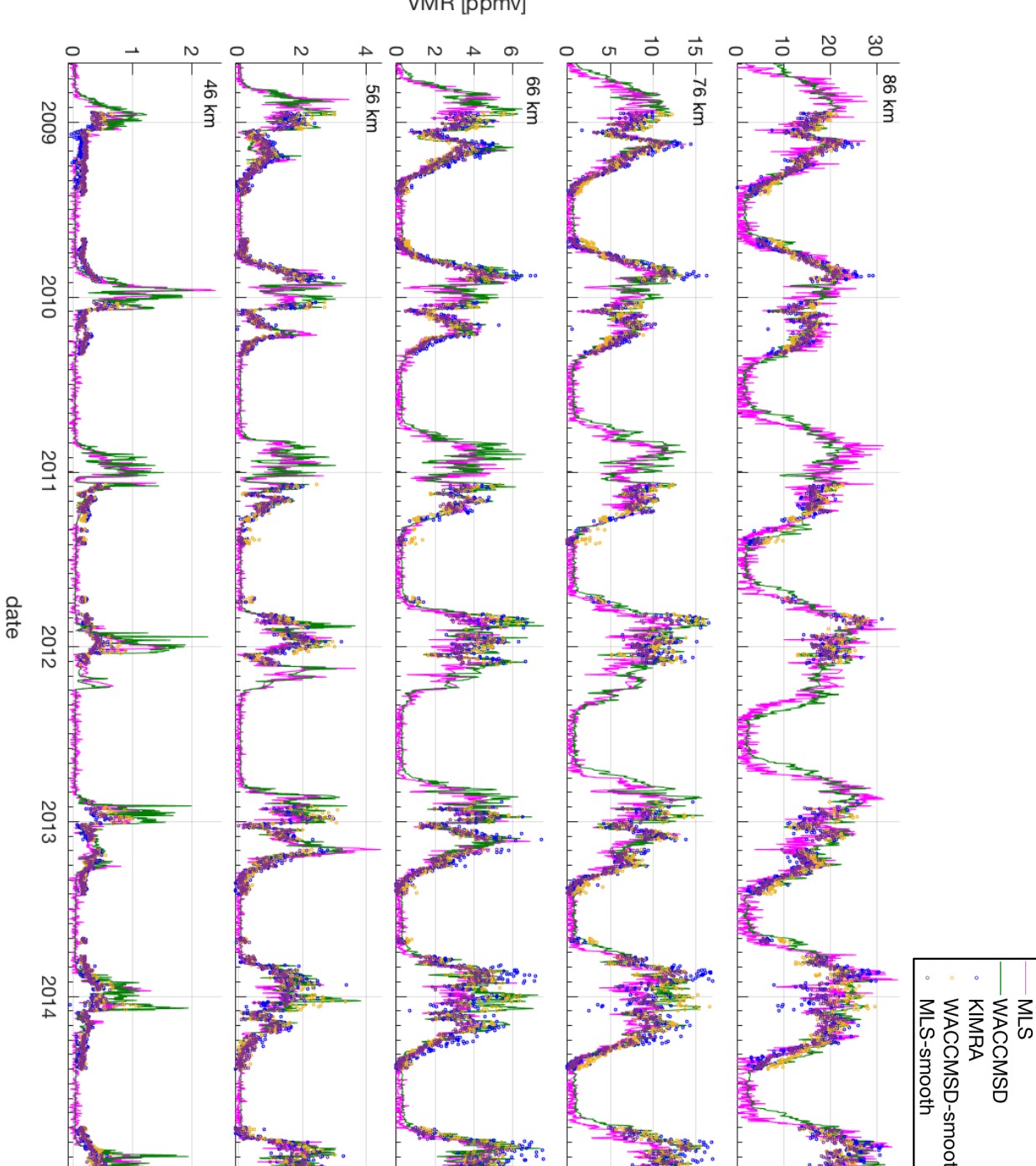

**Figure 1: Comparisons of daily CO VMRs from KIMRA, MLS, and SD-WACCM above Kiruna for 2008 through 2014. Values are displayed at 46, 56, 66, 76, and 86 km altitude. Correlation and regression coefficients for the datasets are given in Table 2. See Sect. 2.4 for details.**

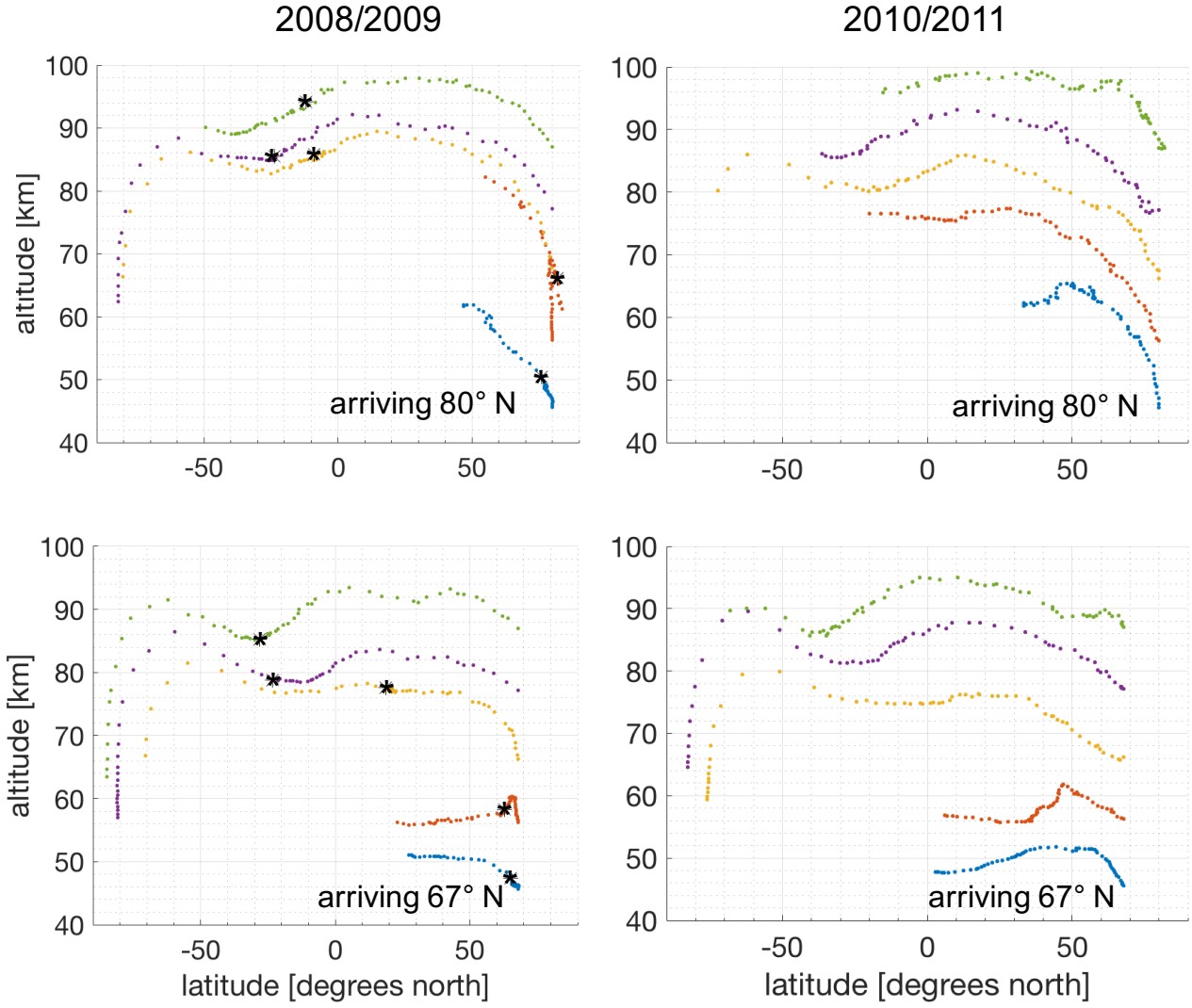

**Figure 2: Trajectories of air parcels advected backwards 60 days in time from February 28th, 2009 and 2011, by the TEM circulation using SD-WACCM. Trajectories are calculated for air parcels arriving at 67° N and 80° N, at 46, 56, 66, 76, and 86 km altitude and locations are plotted at midnight each day. Parcel positions on January 28th 2009 (start date of an SSW) are indicated with black asterix. See Sect. 3 for more details.**

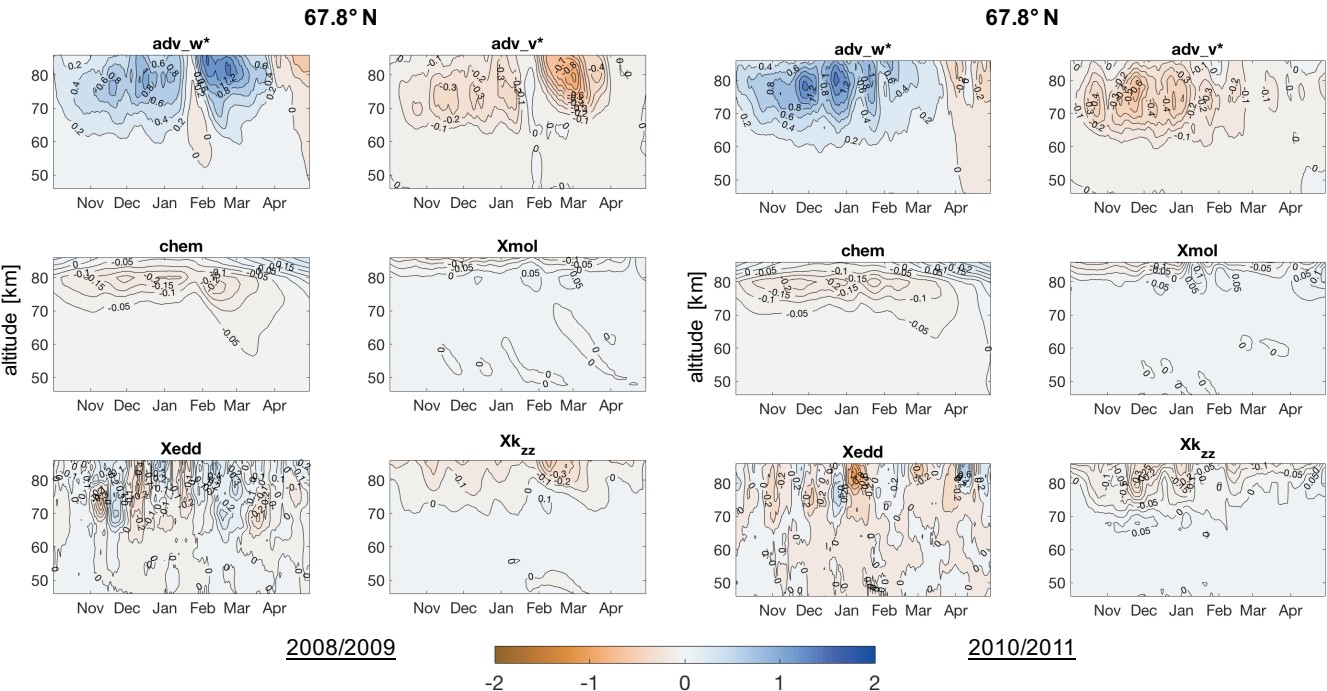

**Figure 3: 11-day running mean tendencies of CO (in ppmv/day), calculated using daily averaged SD-WACCM output. Tendencies shown are for 67.8° N for the winters of 2008/2009 and 2010/2011. See section 3.1 for a description of the tendencies, which are represented in the TEM continuity equation. Note that the labelled contours may be differently spaced in each panel, to match the range of values for individual tendencies.**

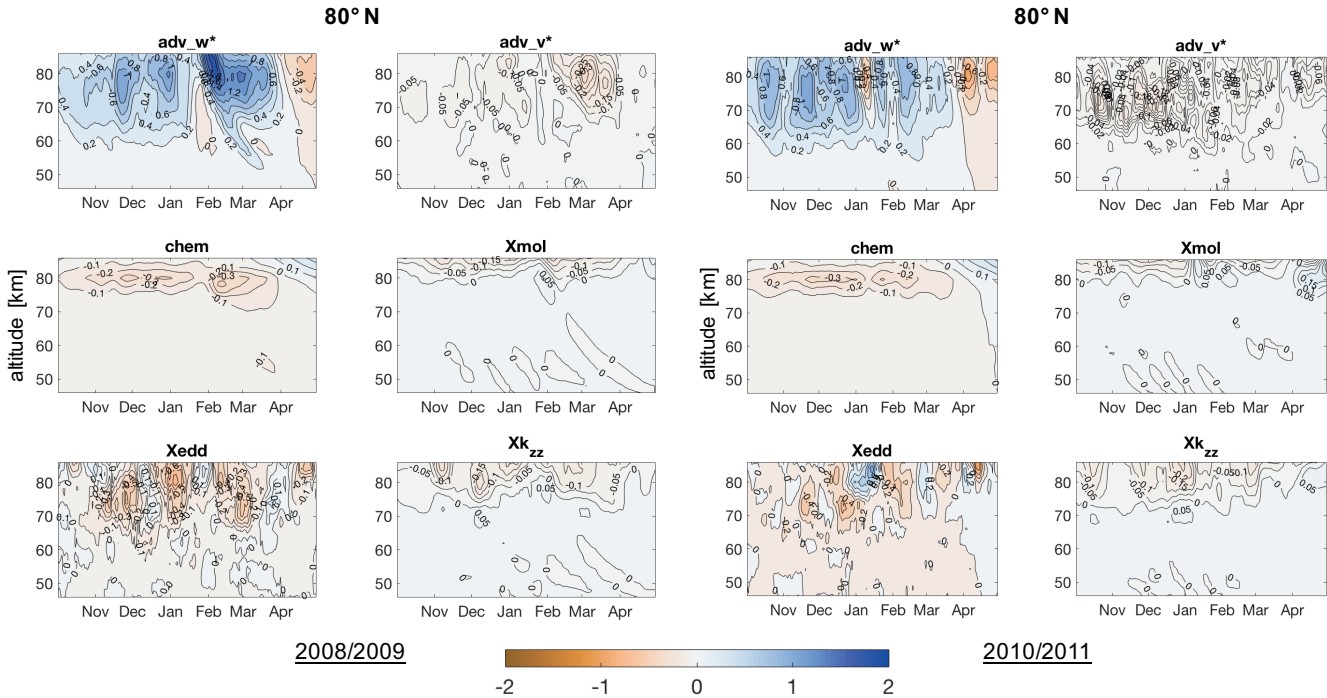

**Figure 4: The same as Fig. 3, but for 80° N.**

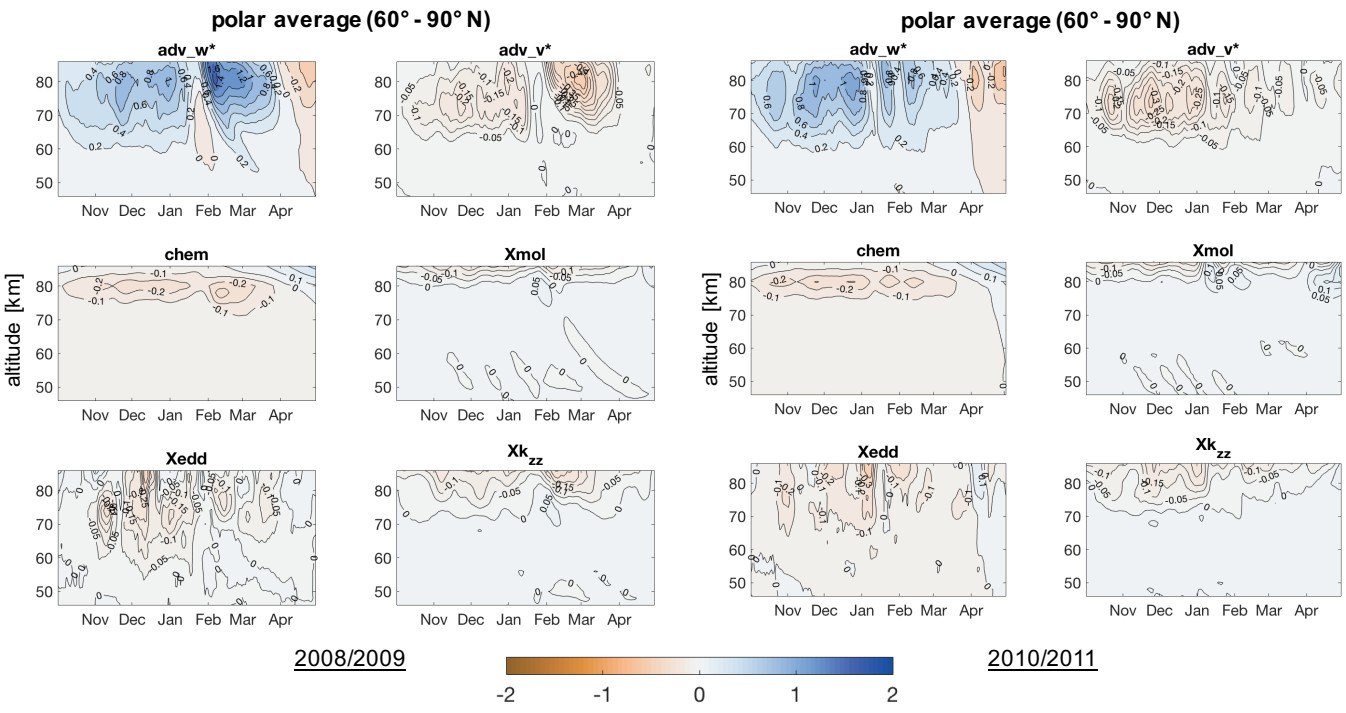

**Figure 5: The same as Fig. 3, but for the north polar average (60° − 90° N).**

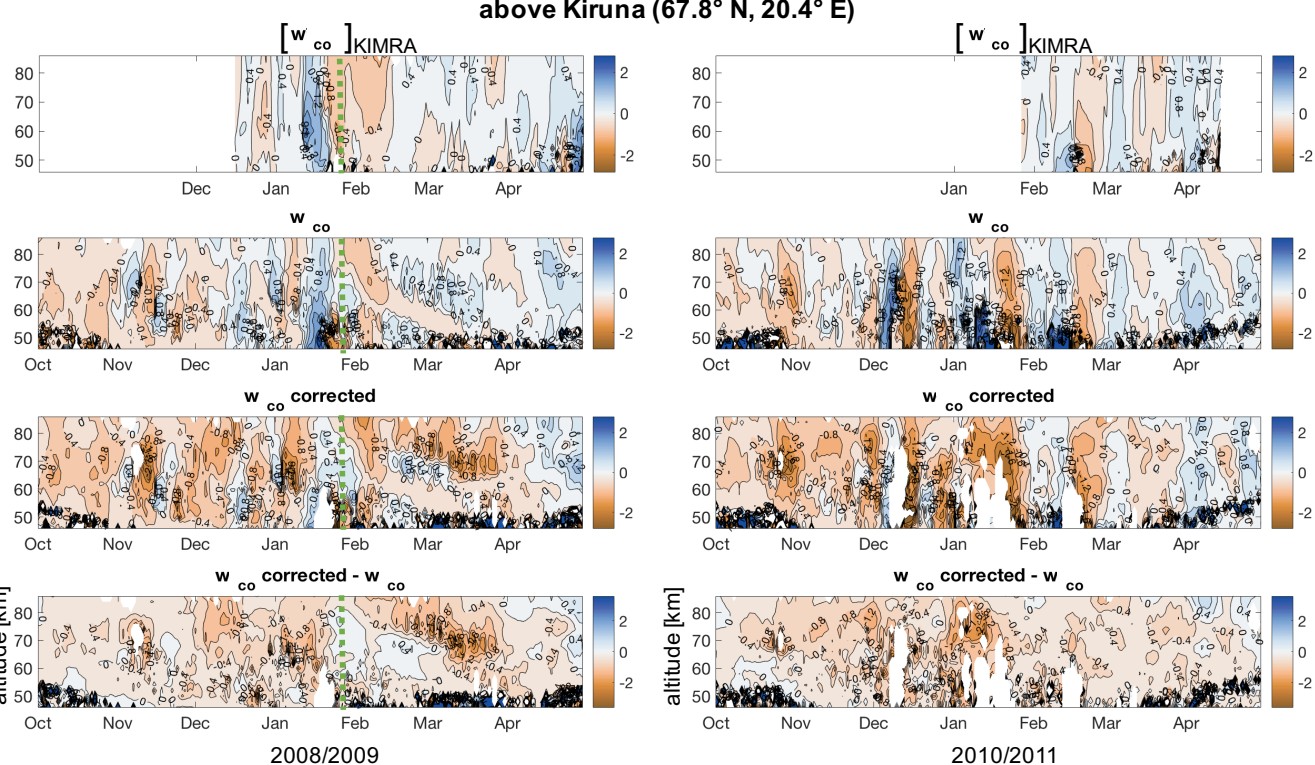

**Figure 6: Rates of vertical motion, in km/day, calculated by tracking CO VMRs over time. $w_{CO}$ is calculated using daily averaged SD-WACCM CO profiles. $w_{CO}$ *corrected* is calculated using a combination of SD-WACCM CO profiles and TEM tendencies (see Sect. 4 for details). The difference between the two rates of descent is also shown. The results plotted are for above Kiruna for the winters of 2008/2009 and 2010/2011. Contour lines are spaced by 0.2 km/day. Areas with tightly packed contours (black areas) occur when there are very low CO VMRs and the calculation method is unreliable. White areas are where a CO VMR could not be tracked within the shown altitude range. The start date of the SSW on January 28th 2009, is shown with a vertical green dashed line. Rates calculated using KIMRA CO data are also included in the upper panel, titled $[w_{CO}]_{KIMRA}$.**

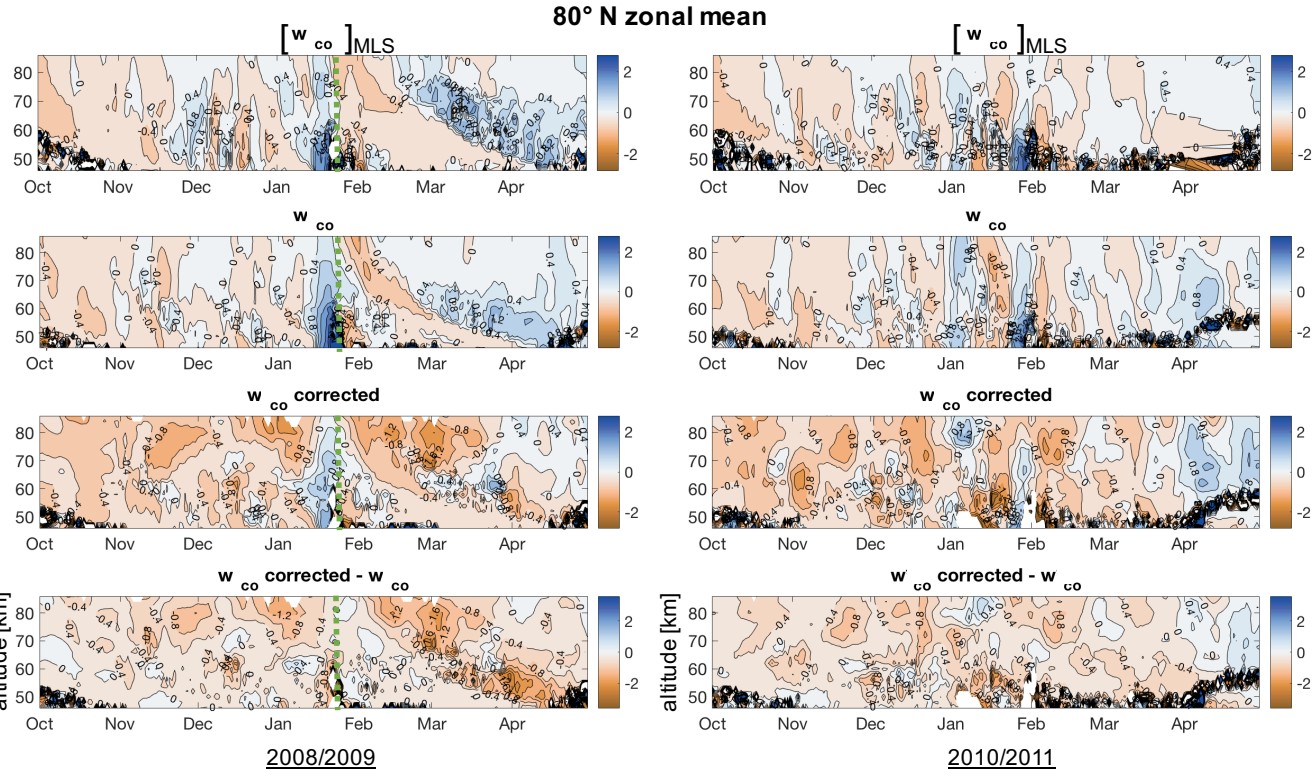

**Figure 7: The same as Fig. 6, but for a zonal mean at 80° N, and including rates calculated using MLS CO data in the upper panel, titled $[w_{CO}]_{MLS}$.**

**polar average (60° - 90° N)**

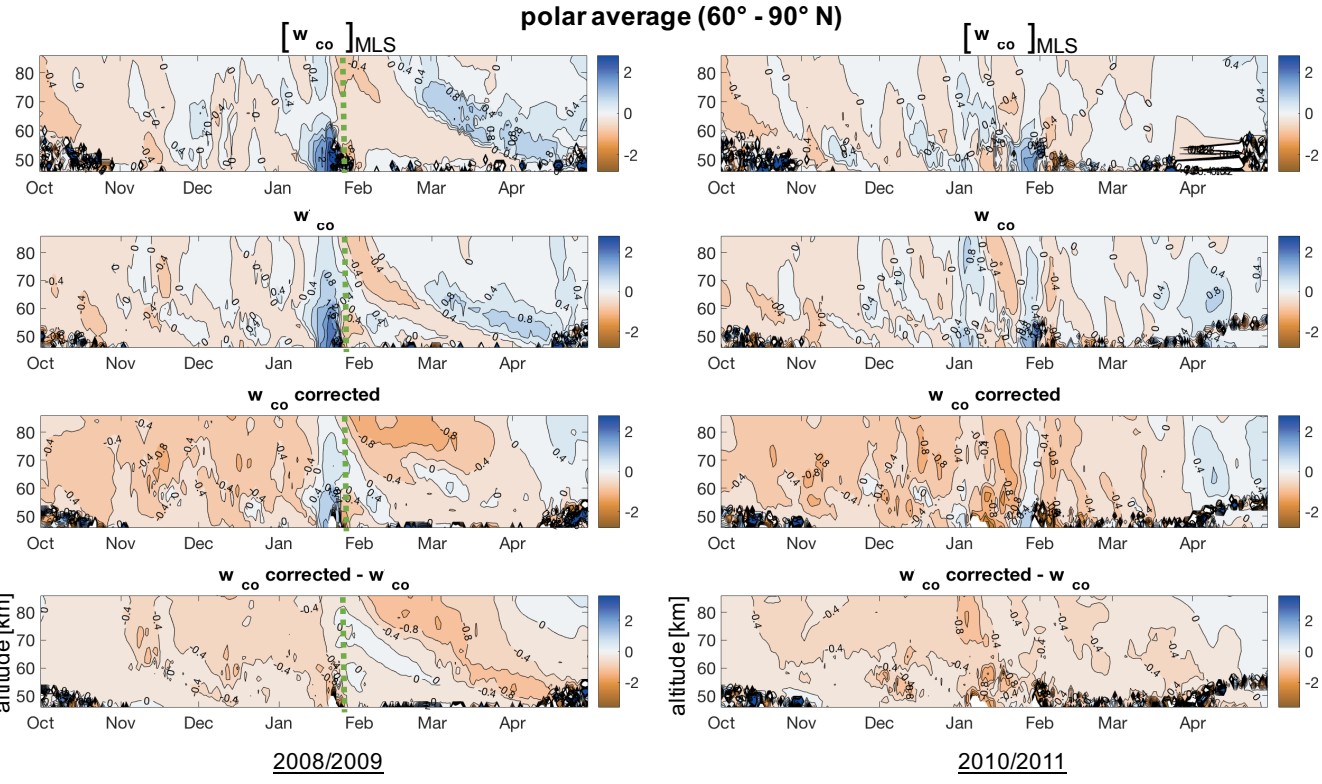

Figure 8: The same as Fig. 6, but for the north polar average (60° − 90° N), and including rates calculated using MLS CO data in the upper panel, titled $[w_{CO}]_{MLS}$.

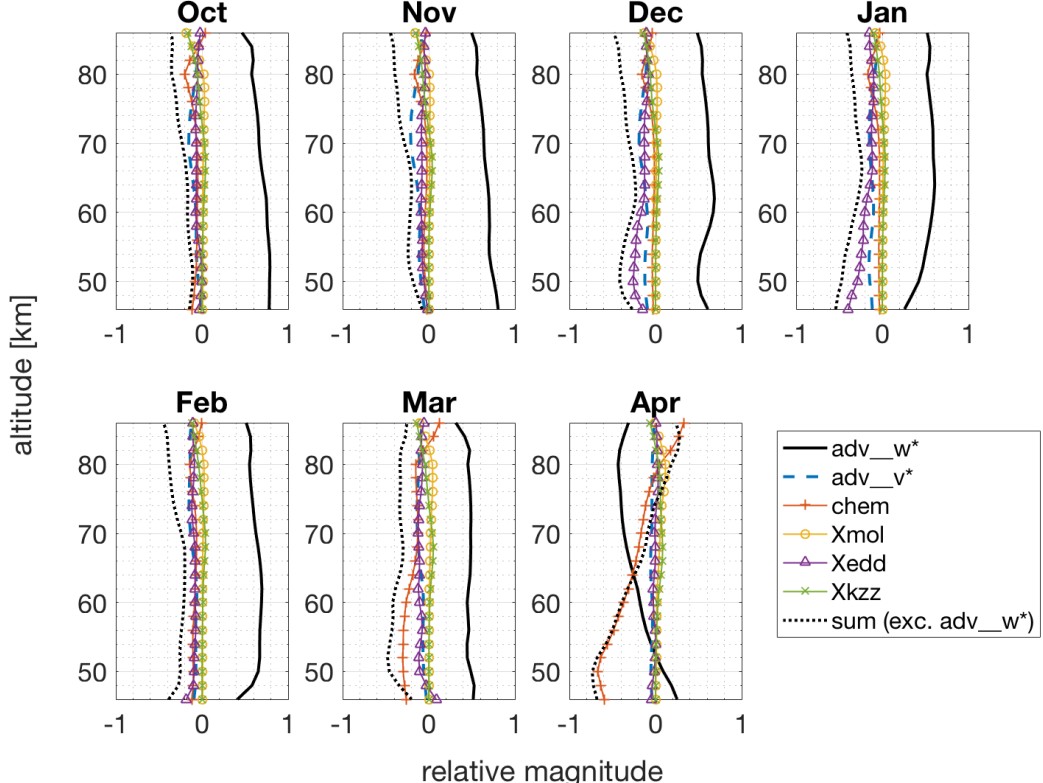

**Figure 9: North polar average (60° – 90° N) monthly mean CO tendencies (see Eq. (2)) around winter time, calculated using SD-WACCM output from January 2008 to April 2014. Model output corresponding to 10 days directly before and after a SSW, as well as the start-date, are excluded. The values are expressed relative to the monthly means of the sum of the absolute values of all tendencies at a given altitude, and are referred to as relative strengths (see Sect. 5). A relative strength of the sum of the tendencies, excluding $adv\_w*$, is also plotted.**

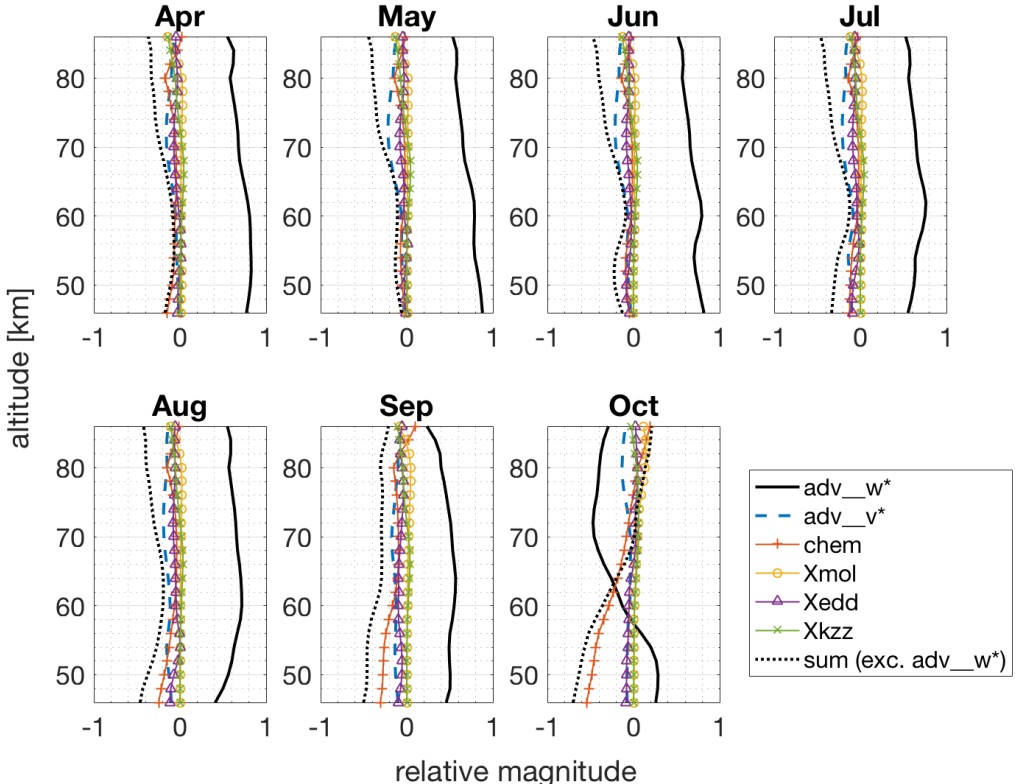

**Figure 10: Same plots as for Fig. 9, but for the south polar average (60° – 90° S), around the time of southern hemisphere winter, and with no exclusion of model output for SSWs.**

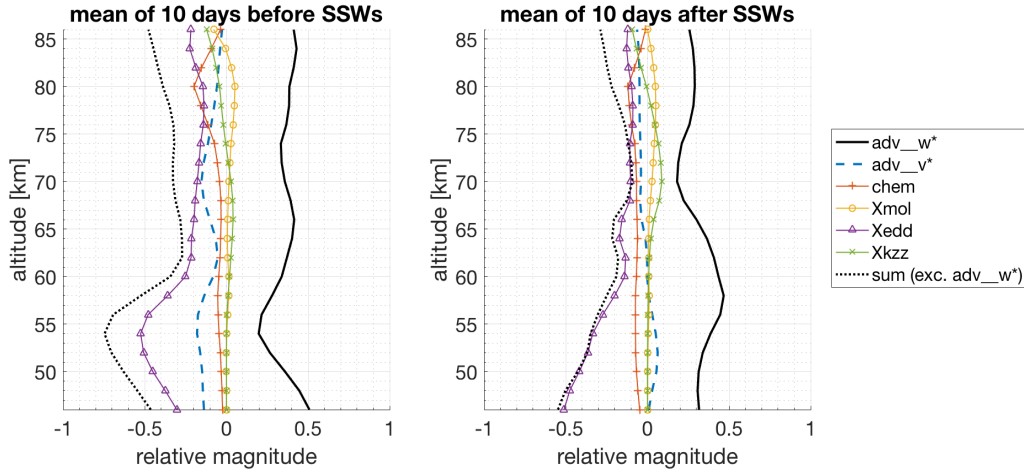

**Figure 11:** Same as Fig. 9 (60 – 90 N), but for the 10 days directly before and after SSWs within Jan 2008 to April 2014. See Sect. 5 for details.

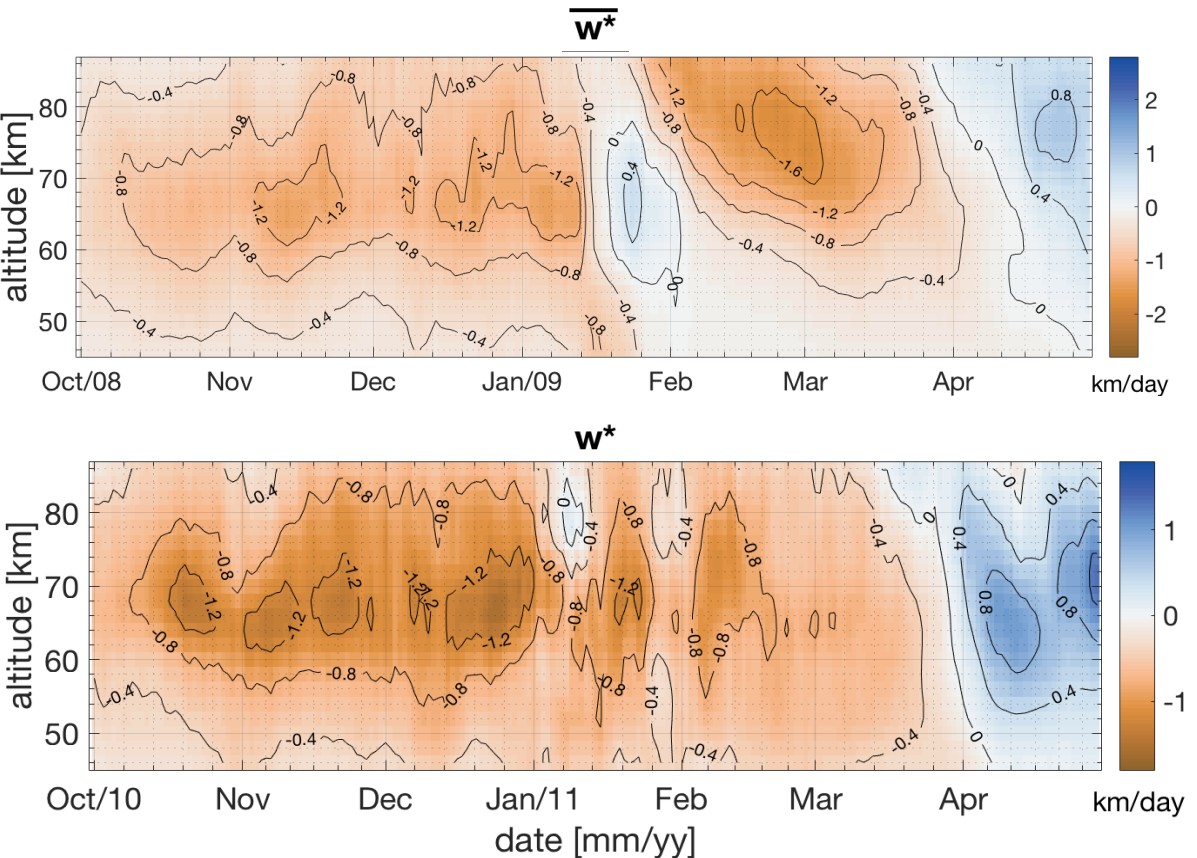