# Peer review of "Assessing the ability to derive rates of polar middle-atmospheric descent using trace gas measurements from remote sensors"

_Atmospheric Chemistry and Physics, 2017_

## Referee Comment (RC1) · Anonymous Referee #1 · 23 Jun 2017

The study significantly contributes to the derivation of the descent rate in the polar middle atmosphere. The authors find that previous studies underestimated the descent rate by a factor of 3 or more. Their data analysis is careful and the good agreement of the SD WACCM CO VMR time series with the observations are good reasons to believe their new finding of a fast descent rate. Thus, the study is appropriate for a publication in ACP and I only suggest some minor corrections.

1) I was not aware that the model descent rate differs so much from the observed descent rate. Straub et al. (2012) found a small descent rate of 325 m/day for the SD WACCM simulation. Please can you argue why SD WACCM is now faster in your

simulation?

2) p.1 line 28 and at other places. I would not use "concentration" since you only work with the volume mixing ratios (VMR). I would introduce VMR in the beginning and then you can always write CO VMR instead of CO concentration.

3) p.3 line 8 it is unclear for me what you mean with a "quiet winter"

4) p.4 line 18 discuss instead of discusses

5) p.7 line 17 "negative" means poleward? I guess the sign depends on the hemisphere and you mean the northern hemisphere?

6) p.7 line 28 Is it Hoffman or Hoffmann like in the Bibliography?

7) p.10 line 6 what do you mean with "fall short"?

8) Conclusions : I am missing a statement that trace gas monitoring by ground-based microwave radiometers in the polar region remains invaluable , e.g., for tuning of the SD WACCM model parameters. Otherwise the paper may give the impression that the observations are useless for derivation of the descent rate.

9) Figure 1 The grey background should be changed by a white background since the contrast is not so good.

10) You may mention somewhere the connection between the polar descent rate and the Brewer-Dobson circulation.

---

## Referee Comment (RC2) · Anonymous Referee #2 · 17 Jul 2017

The authors use results from the WACCM model together with observations of CO from two sensors to investigate how well descent rates can be derived from a chemically (nearly) inert tracer with a strong vertical gradient in the altitude range 45-85 km (i.e., the mesosphere). It is found that considering corrections due to horizontal advection, turbulence, and chemical loss can imply differences in the descent rates derived from CO of more than 1 km/day particularly around strong sudden stratospheric warmings. Credibility is provided by a comparison of the modelled CO to the two observation data sets which generally show a good agreement. Considering that descent rates derived from these methods mostly lie in the range of 100-300 m/day, this is quite a large margin of error. Inert tracers are widely used to derive descent rates in the

polar winter middle atmosphere – not only in the mesosphere to estimate the input of thermospheric tracers, but also in the stratosphere to derive chemical ozone loss rates – and the paper provides an important caveat for these methods. I found the paper very clearly structured and well written, and recommend publication in ACP with a few minor changes.

Page 10, line 12: what does it mean if "w* corrected" derived from modeled CO using corrections from the model itself does not provide the model w*? If equation 1 is a correct description of all terms affecting CO in the model, then "w* corrected" should provide w* in a self-consistent way. I would say that this means that the terms in Equation 1 do not reflect what the model does to CO. I would expect that in the model, the resolved eddy term (Xedd) is not treated separately but as part of the advection scheme; in which case it is counted double if subtracted for derivation of "w* corrected". Does this make sense?

As I understand the term, the middle atmosphere comprised the stratosphere and mesosphere. As you really focus on the mesosphere here (the altitude range from 45-85 km) you might want to change the title of your paper to "polar mesospheric descent".

Page 1, lines 16-17: "The relative importance of vertical advection is lessened . . ." that means that other processes become more important, could you add a sentence which? (Turbulence, horizontal advection,. . .?)

Page 1, lines 25 and following: dynamical tracers have also been used (quite extensively) to derive stratospheric descent rates: to distinguish chemical ozone loss from dynamical processes.

Page 2, line 2-3: you could also reference Funke et al, 2014a, b; and Funke et al., 2017.

Page 2, line 16: . . . "and photochemical destruction in the upper mesosphere" limits

the altitudes at which it can be used to the stratosphere and lowermost mesosphere.

Page 4, section 2.2, line 6: can you also state the approximate altitude range of MLS (in km)?

Page 4, section 2.3, line 28: What exactly does daily output mean – once per day at a specific global time (a global snapshot with varying solar zenith angle) or at a specific local time (a global snapshot with nearly fixed solar zenith angle), or output of daily averages? For a dynamical tracer this probably does not make a big difference apart from some impact of the tidal phase in the upper mesosphere.

Page 5, lines 1-2, discussion of Figure 1: Figure 1 is too small – in my A4 one page per page printout each panel is about 1 cm high, making it very hard to distinguish the lines. You could more than double the vertical range of the panels without filling the page. Please do.

Page 5, line 7: "... but a systematic change in the results ... isn't found ..." despite the very cramped figure (see my previous comment) I do see a systematic difference between MLS and WACCM in early and late winter, i.e., in the buildup and decrease of the winter maximum: the winter maximum starts earlier and lasts longer in WACCM than in MLS, at least above 66 km.

Page 6, line 2-3: "The Prandtl number is 2 for the model runs in this work" I am not quite clear what this means. My understanding is that the Prandtl number describes a physical property of a gas or liquid, namely the relation between momentum diffusivity and thermal diffusivity; as such it should be an exact quantity. The Prandtl number of gases is usually given as lower than 1; for air, values around 0.7-0.8 are given. Does this change around the mesopause (where molecular diffusion becomes more important) compared to the lower atmosphere, or is this really used as a scalable fudge factor in WACCM? – I am aware that this is a feature of WACCM which has been implemented for a good reason; I'm not suggesting that this is changed. I am just curious what it means.

Page 6, line 5: The terms of Eq. 1 are "renamed" here. "Rewritten" suggests that you adapted the terms mathematically.

Page 6, lines 18 and following: I found it quite intriguing that air parcels ending above 66 km actually have their origin in the summer hemisphere. Maybe you can add a mention of this here.

Page 8, line 17 and following, discussion of Figure 4: again, I was intrigued to see that differences between wco and wco corrected sometimes are larger than 1 km / day: two to ten times larger than (most) estimates of descent rates based on tracers as given in Table 1. That really is a big discrepancy.

Page 10, lines 7-12: here you compare w* from the model (Figure 8) with values derived from tracer observations (Figure 4) – it would certainly be easier to follow your argument here if a) the panels in Figure 4 were larger, and b) more importantly, the scale of the colour bars was the same in Figure 4 and 8. It is difficult to appreciate that the values of "w* corrected" provided by tracer observations in 60-90° (Figure 4) is really smaller than the values provided from model wind fields in Figure 8, as the scale in Figure 8 actually covers a smaller range (-1 to 1 km/day compared to -2 to 2 km/day in Figure 4).

Page 10, line 18: see my comment above – what does a Prandtl number of 2-4 mean?

---

## Referee Comment (RC3) · Anonymous Referee #3 · 21 Jul 2017

The authors use WACCM to show the tendencies in CO for the winters of 2008/2009 and 2010/2011. As the authors show, using WACCM, vertical advection is not the only important contributor to these tendencies. These model results are generally reasonably discussed, although I do have some concerns about the presentations in some of the figures (as detailed below). But my much more serious concern is that the authors fail to make appropriate use of their measurements.

Before comparing descent rates in the models and measurements, and before addressing the six major processes that govern this overall tendency in the model, the authors should first show a comparison of the overall CO tendencies in measurements

and models. Admittedly, the model analysis could continue without such a comparison (as the authors state on page 3), but if the CO tendencies in measurements and models are not similar then why are the measurements included here at all? Such a good comparison of WACCM with the measurement components used in this study would be invaluable in helping to judge the ability of the model to accurately address the issue of descent. However, currently the only figure that shows measured CO is Figure 1, and this is both almost impossible to read and does nothing to help the reader to judge the quality of the data as it relates to this study (I would suggest to remove this figure).

Table 1 – Please put a "+" in front of any rising vertical motions to assure the reader that the "-" has not just accidentally been left out.

Figure 3 – The main point of this figure seems to be that "CO VMRs cannot be attributed solely to vertical advection." To make that point the authors need to put all of the contour plots on the same scale. As is, I'm not even convinced that "Tendencies due to resolved eddy diffusion (ðiŚŃðiŚŠðiŚŚðiŚŚ) are the most variable", since I can't compare Xedd plots with the advection plots. If as a result of using consistent scales some plots are left blank, then it's certainly fine to reduce the number of panels and mention the negligible effect of certain terms in the text. And, as mentioned above, there should, in addition to the current 6 panels, be a panel showing "total CO tendency" from both measurement and model.

The chemistry and the two advection terms in Figure 3 generally seem to peak at  $\sim$ 80km. Is there a physical reason for this (if so please explain) or is this related to changes in the model that occur near this level? In the text there is a comment about a chemical sink layer near this altitude. Secondly, please more explicitly explain the normalization applied to Figure 5.

In the conclusion, and elsewhere, the authors declare that using tracer isolines is "invalid". Yet, if I understand Figures 5 and 6 correctly, there are several months and altitude ranges (e.g. near the winter solstice in the lower mesosphere) where w\* does
seem to be the dominant term. A more nuanced conclusion would therefore seem to be in order.

---

## Referee Comment (RC4) · Anonymous Referee #4 · 1 Aug 2017

First review of Ryan et al. entitled "Assessing the ability to derive rates of polar middle-atmospheric descent using trace gas measurements from remote sensors" for publication in ACP. The authors use the SD-WACCM model to argue that processes other than vertical advection of CO are important in the calculation of polar winter descent rates in the upper stratosphere and mesosphere. The paper is well written and the results will be of interest to the scientific community. I recommend publication after the following revisions.

—General comments— I echo here a comment made by another reviewer that the authors need to first show consistency between the CO measurements and the evolution

of CO in SD-WACCM. If the CO tendencies between the obs and the model do not agree then it's not appropriate to use the obs to calculate w-corrected.

Overall, both in the abstract (maybe even the title), throughout the paper, and in the conclusions, the authors need to emphasize that these results are based on SD-WACCM. Report quantitative error estimates to vertical motions derived from tracers instead of using provocative language like "found to be invalid".

Figures 1, 3, and 4 are too small, bordering on illegible. In many cases all of the figure panels shown are neither introduced nor discussed. Please reduce the content of each of these figures to simply support the key point being made in the text.

Figures 3, 4, and 8: swap the color bar to be blue for negative values and red for positives. Whenever possible, hold fixed the color bar range so that comparisons between winters and between tendency terms can be made.

—Line-by-line comments— Abstract Line 16 - "the relative importance of vertical advection is lessened..." – by how much? Give %

1 Introduction Page 2, lines 25-35: "...defining the edges of the polar vortex is not straightforward." – cite Harvey et al. (2009) and Harvey et al. (2015)

2.1 KIMRA, Page 3, line 29: does "average precision" mean daily average?

2.2. MLS, Page 4 line 7 - Does "highest pressure level" refer to the pressure level at the highest altitude? Line 8 - Does "averaged to produce daily profiles" in some spatial region?

2.3 SD-WACCM – Given the fallibility in MERRA winds (mentioned in the intro) in the upper stratosphere and their inability to properly model the elevated stratopause in February of 2009, what (if any) impact does this have on SD-WACCM and the conclusions?

2.4 CO VMR comparison, Page 5 Figure 1 is inadequate. Please compare the model

and measured CO in a comprehensive way that convincingly demonstrates that CO tendencies are in agreement.

Figure 2 – increase panel size and symbol size. Reword last line of the caption to be "Parcel positions on Jan 28th (start of the 2009 SSW) is indicated by black asterisks."

3.3 Tendencies of CO during Arctic winter Page 6, line 28 – "...in depth analysis is not made as it is not the focus of the study." – Then can the results be summarized in fewer than 36 panels? Page 6, line 31 – "...decrease in CO VMRs" add "in the upper mesosphere" – Does this mean air is ascending there? Page 7, first paragraph – Mention different color scales. Give relative magnitudes wrt w*, i.e., chem is 10% of w*. Can we interpret negatives = ascent/poleward and positives = descent/equatorward (or is it not that simple)? Page 7, first paragraph, line 7 – "...because of proximity to the edge of the polar vortex." – No, both are well inside the vortex core. Page 7, line 14 – "There is also a brief change to a positive" – do you mean negative? Page 7, lines 19-20 – This is very useful. Please do this for all tendency terms.

4 Rates of vertical motion with KIMRA and MLS Page 7, lines 30-31 – "...the concentration is adjusted using the tendencies of the continuity equation..." add "from WACCM". This section will gain credibility after showing that the model and observed tendencies are in agreement. Page 8, lines 17-20 – please reword. Are derived descent rates stronger because they need to counteract the opposing terms? Page 8, line 23 – "...around the time of SSW is decreased..." could use an arrow in the figure to highlight this location.

Figure 4 – can any of this information be shown using a scatter plot with w* along one axis and w*-corrected along another? The points could be colored by altitude. It is currently very difficult to look at the panels and understand the comparisons quantitatively. How often would you get points with opposite signs? Is it more likely to get different directions up high or down low?

Figure 5, 6, and Page 9 – refer to 60-90 as the "polar cap" – not the pole.

6 Discussion, Page 10 line 2 - "...not a valid one." Add "according to SD-WACCM" line 30 – Reword "...are representative for the complete mesospheric air..."

7 Conclusion, Page 11, line 17 – "...no months during polar winter when vertical mean advection dominates the budget of CO to such an extent that vertical mean velocity can be accurately derived." – This statement is too strong. Instead, give error estimates as a function of altitude and time over the winter.

---

## Author Comment (AC1) · 15 Sep 2017

The authors would like to take the opportunity to thank the reviewers for their comments and for taking the time to offer them. We believe the manuscript has been improved with the helpful input.

**Response to Reviewer 1**

The study significantly contributes to the derivation of the descent rate in the polar middle atmosphere. The authors find that previous studies underestimated the descent rate by a factor of 3 or more. Their data analysis is careful and the good agreement of the SD WACCM CO VMR time series with the observations are good reasons to believe their new finding of a fast descent rate. Thus, the study is appropriate for a publication in ACP and I only suggest some minor corrections.

1) I was not aware that the model descent rate differs so much from the observed descent rate. Straub et al. (2012) found a small descent rate of 325 m/day for the SD WACCM simulation. Please can you argue why SD WACCM is now faster in your simulation?

The value quoted by Straub et al. (2012) is for an average of the SD-WACCM w\* over the altitude range of 0.6 hPa to 0.06 hPa (approx. 52 – 68 km), and over the time from 5 February to 5 March. This is to match the averaging from the ground-based instrument. The averaging information for the instrument is now included in the edited Table 1 of the new manuscript. Straub et al. (2012) also show the time-averaged profile of WACCM-SD w\* with values ranging from 150 – 700/800 m/day. The daily values of w\* and another trajectory analysis reach 1200 m/s at 0.06 hPa.

Tabe 1 has been edited to include information about the altitude ranges over which the descent rates were calculated and a whether averaging was used. Section 1 also now contains the following information:

"The altitude range over which the rates were determined, and whether averaging was performed, is also shown in Table 1. It is important to note that an average over altitude can mask the higher descent rates that are found in the mesosphere. For example, Straub et al. (2012) show a descent rate of -325 m/day from averaged modelled wind profiles, between 0.6 hPa (~52 km) and 0.06 hPa (~68 km), whereas the individual wind profiles often show descent rates larger than -1000 m/day at 0.06 hPa."

2) p.1 line 28 and at other places. I would not use "concentration" since you only work with the volume mixing ratios (VMR). I would introduce VMR in the beginning and then you can always write CO VMR instead of CO concentration. This has been done.

3) p.3 line 8 it is unclear for me what you mean with a "quiet winter" This has been changed to say "a winter with a relatively stable vortex"

4) p.4 line 18 discuss instead of discusses

This has been fixed.

**5) p.7 line 17 "negative" means poleward? I guess the sign depends on the hemisphere and you mean the northern hemisphere?**

The lines have been edited to clarify that the "negative" refers to the CO tendency, and the direction of  $v^*$  is towards the winter pole. "This is expected, considering the direction of  $\overline{v^*}$ , toward the winter pole, and the low-to-high gradient of CO from lower to higher latitudes in the winter hemisphere"

**6) p.7 line 28 ls it Hoffman or Hoffmann like in the Bibliography?**

It should be Hoffmann. This has been fixed.

**7) p.10 line 6 what do you mean with "fall short"? This has been edited to *"are smaller than"**

8) Conclusions : I am missing a statement that trace gas monitoring by ground-based microwave radiometers in the polar region remains invaluable , e.g., for tuning of the SD WACCM model parameters. Otherwise the paper may give the impression that the observations are useless for derivation of the descent rate.

A line has been added to the end of the conclusion: "Continuous ground-based and satellite measurements of trace gases remain an essential tool in understanding the short- and long-term evolution of the middle atmosphere, as well as for the validation and parameterisation of atmospheric models."

**9) Figure 1 The grey background should be changed by a white background since the contrast is not so good.**

The figure has been edited to have a white background. The panels were edited and made larger to make the data more readable. The layout has been changed to landscape.

**10)** You may mention somewhere the connection between the polar descent rate and the Brewer-Dobson circulation.**

Section 3.2 has been edited and now contains the statements: "The magnitude of the TEM wind is larger for the higher altitudes, as is also shown in Smith et al. (2011), and the air parcels that arrive above 66 km altitude originate in the summer hemisphere. The parcels that arrive below this, which could be considered as part of the Brewer Dobson circulation (Brewer, 1949), originate at latitudes closer to the equator."

---

## Author Comment (AC2) · 15 Sep 2017

The authors would like to take the opportunity to thank the reviewers for their comments and for taking the time to offer them. We believe the manuscript has been improved with the helpful input.

**Response to Reviewer 2**

**The authors use results from the WACCM model together with observations of CO from two sensors to investigate how well descent rates can be derived from a chemically (nearly) inert tracer with a strong vertical gradient in the altitude range 45-85 km (i.e., the mesosphere). It is found that considering corrections due to horizontal advection, turbulence, and chemical loss can imply differences in the descent rates derived from CO of more than 1 km/day particularly around strong sudden stratospheric warmings. Credibility is provided by a comparison of the modelled CO to the two observation data sets which generally show a good agreement. Considering that descent rates derived from these methods mostly lie in the range of 100-300 m/day, this is quite a large margin of error. Inert tracers are widely used to derive descent rates in the polar winter middle atmosphere – not only in the mesosphere to estimate the input of thermospheric tracers, but also in the stratosphere to derive chemical ozone loss rates – and the paper provides an important caveat for these methods. I found the paper very clearly structured and well written, and recommend publication in ACP with a few minor changes.**

**Page 10, line 12: what does it mean if "w* corrected" derived from modeled CO using corrections from the model itself does not provide the model w*? If equation 1 is a correct description of all terms affecting CO in the model, then "w* corrected" should provide w* in a self-consistent way. I would say that this means that the terms in Equation 1 do not reflect what the model does to CO. I would expect that in the model, the resolved eddy term (Xedd) is not treated separately but as part of the advection scheme; in which case it is counted double if subtracted for derivation of "w* corrected". Does this make sense?**

The terms in equation 1 do represent the changes in CO VMR, but in the Transformed Eulerian Mean (TEM) representation of the atmosphere. It is correct to say that what the model "does" in simulating the atmosphere is not the same as following the TEM equations. Equation 1 is a highly derived equation (Andrews et al., 1987), the terms of which are calculated using the output from SD-WACCM.

The TEM offers a way to represent the atmosphere as an interaction of a mean flow with disturbances, i.e., eddies and waves, imposed upon it, and it contains a description of a mean meridional flow in the atmosphere. The interaction is generally a two-way process and the disturbances feed back into the mean flow through non-linear effects.

So the calculation of w*, using the TEM formalism and using CO VMRs, will have some differences due to the way that each is calculated. The calculation of w*_corrected is also somewhat crude (as mentioned in the paper and now expanded upon in the new manuscript), as it combines tendencies in the TEM formalism with values derived using CO VMRs in the atmosphere. These points are now emphasized in Section 4 and it is made clearer that the goal

of w*_corrected is to get a qualitative estimate of the errors that may be incurred by assuming pure vertical advection when using tracers to calculate w*.

*"The resulting rate is called $w_{CO}$ corrected. This could be considered a crude approach, combining daily averaged CO output with CO tendencies calculated using the TEM formalism, but the aim here is to provide an estimate of the errors that may be incurred by neglecting influences on CO other than vertical advection. In any case, the results involving $w_{CO}$ corrected are discussed in a qualitative manner, instead of for quantitative error analysis."*

The discussion in Section 6 offers some other reasons for the discrepancy: mainly parameterization of gravity waves in the model, which play a role in the strength of w* and also in the parameterised eddy flux divergence (Xkzz), and the time resolution of the model output. Meraner and Schmidt (2016) found differences in calculated w* when using 6-hourly average model output compared to daily average output.

**As I understand the term, the middle atmosphere comprised the stratosphere and mesosphere. As you really focus on the mesosphere here (the altitude range from 45-85 km) you might want to change the title of your paper to "polar mesospheric descent".**
The term was chosen because altitude of the stratopause can be around 55 km, which would mean that a 10 km layer of the analysis is within the stratosphere.
Because the analysis does not cover the whole of the middle atmosphere, the first line of the abstract has been edited to express the altitude range. Combined with the title, the reader will now immediately know the area under investigation: *"We investigate the reliability of using trace gas measurements from remote sensing instruments to infer polar atmospheric descent rates during winter in the 46 – 86 km altitude range."*

The altitude range is explicitly mentioned in the conclusions now also.

**Page 1, lines 16-17: "The relative importance of vertical advection is lessened : : :" that means that other processes become more important, could you add a sentence which? (Turbulence, horizontal advection,: …?)**
The sentence has been edited to read:
*"The relative importance of vertical advection is lessened during periods directly before and after a sudden stratospheric warming, mainly due to an increase in eddy transport"*

**Page 1, lines 25 and following: dynamical tracers have also been used (quite extensively) to derive stratospheric descent rates: to distinguish chemical ozone loss from dynamical processes.**
The following statement and references have been added to Section 1: *"This technique has often been used in combination with ozone measurements to separate chemical and dynamical influences when determining ozone depletion (e.g. Proffit et al., 1990, 1993; Müller et al., 1996, 2003; Salawitch et al., 2002; Rösevall et al., 2007)."*

**Page 2, line 2-3: you could also reference Funke et al, 2014a, b; and Funke et al., 2017.**

These references have been added to the main text and to the references section.

**Page 2, line 16: … "and photochemical destruction in the upper mesosphere" limits the altitudes at which it can be used to the stratosphere and lowermost mesosphere.**
We would prefer not to provide such a definition because the referenced paper (Lee et al., 2011) does not make such a definite statement.
Lee et al. (2011) states: "However, it is not as good a tracer as CO for diagnosing vertical motions throughout the middle atmosphere due to the relative complexity of photochemical sources and sinks in the stratosphere."

**Page 4, section 2.2, line 6: can you also state the approximate altitude range of MLS (in km)?**
The sentence in question now reads: *"These CO profiles cover a pressure range of 215 – 0.0046 hPa (approximately 11 – 86 km) …"*

**Page 4, section 2.3, line 28: What exactly does daily output mean – once per day at a specific global time (a global snapshot with varying solar zenith angle) or at a specific local time (a global snapshot with nearly fixed solar zenith angle), or output of daily averages? For a dynamical tracer this probably does not make a big difference apart from some impact of the tidal phase in the upper mesosphere.**
The model output Is daily averages. This sentence has been edited to read: *"Model output of daily averages from 2008 to 2014 are used for this study."*

**Page 5, lines 1-2, discussion of Figure 1: Figure 1 is too small – in my A4 one page per page printout each panel is about 1 cm high, making it very hard to distinguish the lines. You could more than double the vertical range of the panels without filling the page. Please do.**
The panels have been enlarged, and superfluous date labels have been removed to give more space for the panels. The figure has also been changed to landscape layout so that the panels can be made larger. The background was also changed to a light colour at the request of Reviewer #1.

**Page 5, line 7: "; … but a systematic change in the results … isn't found …" despite the very cramped figure (see my previous comment) I do see a systematic difference between MLS and WACCM in early and late winter, i.e., in the buildup and decrease of the winter maximum: the winter maximum starts earlier and lasts longer in WACCM than in MLS, at least above 66 km.**
That particular sentence regards the difference of using bilinear interpolation of the model data, or not.
However, your point about the systematic difference between the model and data remains valid. The following lines have been added to the section to make the point: *"A systematic difference is evident between MLS and SD-WACCM during the times of year when CO VMRs are rapidly increasing or decreasing, with SD-WACCM showing larger values of CO. The difference is*

*most pronounced at higher altitudes and is predominantly during August/September and April/May."*

**Page 6, line 2-3: "The Prandtl number is 2 for the model runs in this work" I am not quite clear what this means. My understanding is that the Prandtl number describes a physical property of a gas or liquid, namely the relation between momentum diffusivity and thermal diffusivity; as such it should be an exact quantity. The Prandtl number of gases is usually given as lower than 1; for air, values around 0.7-0.8 are given. Does this change around the mesopause (where molecular diffusion becomes more important) compared to the lower atmosphere, or is this really used as a scalable fudge factor in WACCM? – I am aware that this is a feature of WACCM which has been implemented for a good reason; I'm not suggesting that this is changed. I am just curious what it means.**

The Prandtl number (Pr) as used in GW parameterizations describes the ratio of momentum flux to heat flux, and is properly thought of as a "turbulent Prandtl number"; in particular, it is not a property of a gas or liquid, but of the process whereby gravity waves dissipate when they "break".

For breaking gravity waves, we really do not know what Pr should be, so to some extent it has been used as an adjustable parameter in GW parameterizations. This is the way Pr is used in WACCM and, in fact, in all models that parameterize GW breaking and attempt to derive turbulent mixing due to such breaking.

Section 3.1 in the edited manuscript now clarifies this with the following sentences: *"The value of $K_{zz}$ calculated with SD-WACCM depends, among other things, upon the Prandtl number (or more properly, the "turbulent Prandtl number"), which describes the ratio of momentum flux to heat flux. The Prandtl number is a property of the process whereby gravity waves dissipate when they "break" (see e.g., Fritts and Dunkerton, 1985, for a more details). The Prandtl number is 2 for the model runs in this work (see Sect. 6) and is used in SD-WACCM to parameterise gravity wave breaking (Garcia et al., 2007)."*

**Page 6, line 5: The terms of Eq. 1 are "renamed" here. "Rewritten" suggests that you adapted the terms mathematically.**

This has been changed.

**Page 6, lines 18 and following: I found it quite intriguing that air parcels ending above 66 km actually have their origin in the summer hemisphere. Maybe you can add a mention of this here.**

Section 3.2 has been edited and now contains the following information: *"The magnitude of the TEM wind is larger for the higher altitudes, as also shown in Smith et al. (2011), and the air parcels that arrive above 66 km altitude originate in the summer hemisphere. The parcels that arrive below this, which could be considered as part of the Brewer Dobson circulation (Brewer, 1949), originate at latitudes closer to the equator."*

**Page 8, line 17 and following, discussion of Figure 4: again, I was intrigued to see that differences between wco and wco corrected sometimes are larger than 1 km / day: two**

**to ten times larger than (most) estimates of descent rates based on tracers as given in Table 1. That really is a big discrepancy.**
The differences between w* and w*_corrected can be quite large, but it is hard to compare them quantitatively to the values listed in Table 1. There are two points here:

The first point is that the differences between w* and w*_corrected were used to provide more of a qualitative estimation of the errors that can be incurred by neglecting processes other than vertical advection. This is mainly due to the crudeness of combining information derived in two different ways: changes in daily averaged CO from the model/instrument, and TEM tendencies. Section 4 of the edited manuscript now clarifies this point:
*"The resulting rate is called $w_{CO}$ corrected. This could be considered a crude approach, combining daily averaged CO output with CO tendencies calculated using the TEM formalism, but the aim here is to provide an estimate of the errors that may be incurred by neglecting influences on CO other than vertical advection. In any case, the results involving $w_{CO}$ corrected are discussed in a qualitative manner, instead of for quantitative error analysis."*

The second point is that the values in Table 1 come from a variety of analyses, some of which are averages in space, time, or both. Averages in altitude generally provide lower values for w* than those that are seen at altitudes above about 70 km.
Straub et al. (2012), for example, quote a value of w* of ~ 325 m/day, from instrument, model, and trajectory analysis. The value is an average over Feb/March and also over an altitude range of 0.6 hPa - .06 hPa (approx. 52 – 68 km). The modelled and trajectory analysis w* profiles, from which the averages are made, often show values of ~1200 m/s at 0.06 hPa.

Table 1 has been edited to state where averaging has been performed, and also the altitude ranges that were used in the studies. A point is now made in Section 1 about noting the effect different averaging techniques.
*"The altitude range over which the rates were determined, and whether averaging was performed, is also shown in Table 1. It is important to note that an average over altitude can mask the higher descent rates that are found in the mesosphere. For example, Straub et al. (2012) show a descent rate of -325 m/day from averaged modelled wind profiles, between 0.6 hPa (~52 km) and 0.06 hPa (~68 km), whereas the individual wind profiles often show descent rates larger than -1000 m/day at 0.06 hPa.*
*."*

**Page 10, lines 7-12: here you compare w* from the model (Figure 8) with values derived from tracer observations (Figure 4) – it would certainly be easier to follow your argument here if a) the panels in Figure 4 were larger, and b) more importantly, the scale of the colour bars was the same in Figure 4 and 8. It is difficult to appreciate that the values of "w* corrected" provided by tracer observations in 60-90 (Figure 4) is really smaller than the values provided from model wind fields in Figure 8, as the scale in Figure 8 actually covers a smaller range (-1 to 1 km/day compared to -2 to 2 km/day in Figure 4).**

Figure 8 from the current manuscript has been edited to have the same colour limits as Figure 4. The scenarios from Figure 4 (spot, zonal mean, and polar mean) have been split into separate figures and the panels made larger so that the plots are clearer.

**Page 10, line 18: see my comment above – what does a Prandtl number of 2-4 mean?**
This is addressed above in response to the referenced comment.

---

## Author Comment (AC3) · 15 Sep 2017

The authors would like to take the opportunity to thank the reviewers for their comments and for taking the time to offer them. We believe the manuscript has been improved with the helpful input.

The authors use WACCM to show the tendencies in CO for the winters of 2008/2009 and 2010/2011. As the authors show, using WACCM, vertical advection is not the only important contributor to these tendencies. These model results are generally reasonably discussed, although I do have some concerns about the presentations in some of the figures (as detailed below). But my much more serious concern is that the authors fail to make appropriate use of their measurements.

Before comparing descent rates in the models and measurements, and before addressing the six major processes that govern this overall tendency in the model, the authors should first show a comparison of the overall CO tendencies in measurements and models. Admittedly, the model analysis could continue without such a comparison (as the authors state on page 3), but if the CO tendencies in measurements and models are not similar then why are the measurements included here at all? Such a good comparison of WACCM with the measurement components used in this study would be invaluable in helping to judge the ability of the model to accurately address the issue of descent. However, currently the only figure that shows measured CO is Figure 1, and this is both almost impossible to read and does nothing to help the reader to judge the quality of the data as it relates to this study (I would suggest to remove this figure). The aim of the paper is to use the individual tendencies of CO in the atmosphere to ascertain whether vertical advection can be considered dominant to such an extent that the atmospheric descent rates can be calculated by observing tracer motion. This cannot be achieved with only measurements, as there is not enough information to separate the contributions to the observed VMRs. A model that simulates accurately the observed evolution of atmospheric CO VMRs is used to separate the contributions, or tendencies.

The comparison of atmospheric CO measured by instruments and modelled with SD-WACCM is vital to determining whether the model accurately represents the CO VMRs that are observed in the atmosphere. Figure 1 has been enlarged, changed to landscape layout, and edited to make the data clearer. Table 2 lists the quantitative results of the comparison and shows the level of agreement between the model and the instruments, which is quite high for daily averages. An in-depth comparison of KIMRA, MLS, and SD-WACCM has been made by Hoffmann et al. (2012a), and the comparison is made in the current manuscript because there have been updates to the model and the datasets. Section 1 now includes this information: *"The values are similar to those found for earlier versions of the model and data (Hoffmann et al., 2012a), with differences mainly due to updates to the modelled CO (Garcia et al., 2014) and the data products (Livesey et al., 2015; Ryan et al., 2017)."*

Section 4 has been edited to include the calculations that were made with the modelled CO, instead of the measured CO. We agree that this is more consistent and avoids the differences between the model and the instruments (Table 2). The results led to the same conclusions

because of the level of agreement between the modelled and measured CO, as stated in the original manuscript.

Section 1 has been edited to clarify the above points:

"The aim of this study is to assess the limits of the above assumption when using tracer measurements from remote sounders to derive rates of vertical motion in the middle atmosphere. Measurements alone do not provide enough information to enable separation of the contributions to changes in tracer VMRs, and so an atmospheric model must be employed. The specified dynamics version of the Whole Atmosphere Community Climate Model (SD-WACCM) is used to determine the relative contributions to changes in CO VMRs during polar winter. The results are combined with daily average modelled CO to estimate the error associated with descent rates calculated assuming pure vertical advection of the tracer. Three commonly used representations of the data are assessed: a local area above a specific location (Kiruna, 67.8° N, 20.4° E, in this case), a zonal mean at a certain latitude (80° N is used as an example), and a polar mean (60° - 90° N). The winters of 2008/2009 and 2010/2011 are used in the study as an example of a winter with a strong SSW and a winter with a relatively stable vortex, respectively. The rate calculations were also performed using CO measurements from the Kirung Microwave Radiometer (KIMRA) and the Microwave Limb Sounder (MLS) (not shown), and the results lead to the same conclusion. This was expected due to the level of agreement found in a comparison of the modelled and measured CO (Sect. 2.4)."

Table 1 – Please put a "+" in front of any rising vertical motions to assure the reader that the "-" has not just accidentally been left out. This has been done.

Figure 3 – The main point of this figure seems to be that "CO VMRs cannot be attributed solely to vertical advection." To make that point the authors need to put all of the contour plots on the same scale. As is, I'm not even convinced that "Tendencies due to resolved eddy diffusion (>`I'S' N>`I 'SŠ.'I'SS') are the most variable", since I can't compare Xedd plots with the advection plots. If as a result of using consistent scales some plots are left blank, then it's certainly fine to reduce the number of panels and mention the negligible effect of certain terms in the text. And, as mentioned above, there should, in addition to the current 6 panels, be a panel showing "total CO tendency" from both measurement and model.

This sentence was supposed to refer to the fact that Xedd varies the most between positive and negative values. It has been edited in the new manuscript to correct this:

"Tendencies due to resolved eddy diffusion (Xedd) show the most variation between positive and negative values, ..."

Figure 3 has been separated, by the locations (i.e., 67N, 80N, and polar average), into three figures so that the panels can be made larger and more easily read.

In originally making the figures, it was decided not to use the same colour range for each tendency because too much of the information is lost from the plots. Because of the varying values of the tendencies (within a winter and from year to year), there are times, for example, when all tendencies show relatively low values. With the same colour bar for each panel, the

information about relative influence is then lost. There are too many instances of this nature to choose a single colour range that includes the relevant information. Instead, it was decided to use both labelled contours and colour bars. While it is admittedly not easy to quickly determine the relative magnitudes of the tendencies from a glance, relevant information is not omitted from the figure. The separation of the Figure 3 into three, makes it easier for the reader to attain this information for each scenario in the edited manuscript.

Section 3.3 now emphasises that there are different colour bars and added contours: "Note that the tendencies are plotted with individual colour scales to retain relevant information when there are low magnitudes, and labelled contours are added."

The chemistry and the two advection terms in Figure 3 generally seem to peak at 80km. Is there a physical reason for this (if so please explain) or is this related to changes in the model that occur near this level? In the text there is a comment about a chemical sink layer near this altitude. Secondly, please more explicitly explain the normalization applied to Figure 5.

The peak around 80 km in the chemical tendency is due to a layer of nighttime OH at this altitude, which acts as a chemical sink for CO. This is stated at the following points of the original manuscript:

Abstract. Page 1, line 16:

"It was also found that CO chemistry cannot be ignored in the mesosphere due to the night-time layer of OH at approximately 80 km altitude."

Section 3.3. Page 7, line 3-8:

"Changes in CO due to chemistry (*chem*) are small below approximately 70 km, but all cases show a sustained sink for CO during the winter in a layer at around 80 km altitude. The layer coincides with the location of a night- time layer of hydroxyl (OH) around 82 km altitude (Brinksma et al., 1998, Pickett et al., 2006, Damiani et al., 2010). OH is known as the dominant chemical sink for middle-atmospheric CO (Solomon et al, 1985)."

Conclusion: Page 11, line 9-12:

"The results also show a chemical sink for CO, present throughout polar night, due to the layer of night-time OH at approximately 80 km."

The higher values of the advection tendencies around 70 to 80 km are mainly due to two points: the first is larger magnitudes of the TEM circulation compared to lower altitudes before there is a turnaround in the direction of the circulation at higher altitudes. The circulation changes from *poleward and downward* to *poleward and upward*. The turnaround point is at approximately 95 km in WACCM.

The second reason is that the vertical gradient of CO, which is proportional to the TEM vertical advection, generally increases with altitude.

The following information is included in the edited manuscript:

"The advection tendencies show maximum values around 70 - 80 km for two main reasons. The first is the larger magnitude of the TEM circulation, compared to lower altitudes, before there is a turnaround in the direction of the circulation at higher altitudes, at which point the circulation changes from poleward and downward to poleward and upward (e.g., Lieberman et al., 2000;

Smith et al., 2011). The turnaround point is at approximately 95 km in WACCM (Smith et al., 2011). The second is the generally increasing vertical gradient of CO with altitude (see Eq. 1)."

The description of the normalisation has been clarified and now reads: "For a given tendency, the daily values are separated by calendar month and averaged, to give a monthly mean tendency. The daily sums of the absolute values of all tendencies are also separated by month and averaged, to give a monthly mean total absolute tendency. The monthly mean tendencies are then normalised by the monthly mean total absolute tendency, and will be referred to here as relative strengths. Using absolute values for normalisation retains the sign of the individual tendencies and avoids a large spread in the results when there is a small denominator (i.e., when the tendencies cancel each other and their sum is near zero)."

**In the conclusion, and elsewhere, the authors declare that using tracer isolines is "invalid". Yet, if I understand Figures 5 and 6 correctly, there are several months and altitude ranges (e.g. near the winter solstice in the lower mesosphere) where w\* does seem to be the dominant term. A more nuanced conclusion would therefore seem to be in order.**

Section 5 has been edited to provide more quantitative information that qualifies the statement in the first sentence of the paragraph, and elsewhere. The variation in the relative strength is discussed and the changes in tendencies with time from earlier is also emphasised: *"There are no months where the relative strength of other processes can be considered negligible compared to the relative strength of adv\_w \*. The closest approximations of this situation are at 50 km altitude in October and at 46 km altitude in November, when other processes contributes 13.7 % and 9.6 % of adv\_w \*, respectively. These percentages then vary significantly with altitude. For October, the value increases to 18.6 % at 46 km, 22.5 % at 60 km, and is 61.13 % at 80 km. For November, the value increases to 34.4 % at 54 km, and is 70.8 % at 80 km.*

The results for the south polar average, in Fig. 6, are qualitatively similar to those for the north. The relative strength of  $adv_w *$  shows a maximum of ~0.8. Both hemispheres show a peak in chem at 80 km for most of winter (see Sect. 3.3). The relative strength of Xedd is not as prominent at the south as the north, likely due to the higher stability of the southern polar vortex. The points at which the relative strength of other processes is smallest compared to  $adv_w *$  are at 56 km in April (8.3 %) and at 46 km in May (6.8 %). For April, the value increases to 22.5 % at 46 km and 21.5 % at 66 km, and is 56.9 % at 80 km. In May, the value increases to 16 % at 54 km, and is 69.1 % at 80 km.

For the 10 days directly before and after SSWs, in Fig. 7, the relative strength of  $adv_w *$  is less than 0.5 at all altitudes. Xedd is strong below 60 km, such that the relative strength of other processes has a larger magnitude than that of  $adv_w *$  at many altitudes. The relative strength of  $adv_w *$  shows a more oscillatory structure with altitude, and there is a local minimum at about 70 km in the data for 10 days after SSWs. There is also a positive peak in the relative strength of  $Xk_{zz}$  after SSWs at this altitude.

Aside from considering what value would classify as negligible, the significant variation in strength of other processes compared to  $adv_w *$ , over altitude, adds complexity to the method of following a tracer over an altitude range to determine the descent rate. One must also

consider that while this section discusses monthly averaged data, tracers are often followed for several days to determine the changes in altitude over that time, and that the magnitudes of each tendency can vary significantly over this time scale (see Figures 3, 4, and 5)."

The referred to statement in Section 6 has been edited to emphasise that the conclusion is indicated by the results, from all sections, using SD-WACCM.

**Section 6:**

"The results of the previous sections, using SD-WACCM, are clear on one indication, that the assumption of observed changes in CO VMRs being solely due to vertical advection is not a valid one."

The referenced sentence in the conclusion has been edited to emphasise that this conclusion is indicated by the results from all sections. Statements of the main results have been added/edited in the conclusion as follows:

- "The results show that dynamical processes other than vertical advection cause non-negligible changes in CO VMRs during winter, and particularly directly before and after sudden stratospheric warmings when eddy transport can become dominant."

- "Significant changes in CO tendencies from SD-WACCM occur on the order of days. The results also show a chemical sink for CO, present throughout polar night, due to the layer of night-time OH at approximately 80 km."

- "Rates of atmospheric motion were calculated when assuming only vertical advection, and corrected rates were calculated by including tendency information for all processes. The differences between the two results are of the same order as the calculated rates, and the rates are prone to showing opposite directions for the mean vertical wind."

- "The "true" rate of atmospheric descent appears to be masked by sinks of CO, and by transport processes that oppose the tendency due to vertical advection"

- "Monthly mean relative tendencies for CO show that the summed magnitude of processes other than vertical advection can constitute a large fraction of the changes in CO VMR. For a given month, the summed magnitude of the other processes, relative to vertical advection, changes by several tens of percent over the altitude range under investigation."

- "The results suggest that there are no months during polar winter when vertical mean advection dominates the budget of CO to such an extent that vertical mean velocity can be accurately derived within the altitude range."

---

## Author Comment (AC4) · 15 Sep 2017

The authors would like to take the opportunity to thank the reviewers for their comments and for taking the time to offer them. We believe the manuscript has been improved with the helpful input.

**Response to Reviewer 4**

**First review of Ryan et al. entitled "Assessing the ability to derive rates of polar middle-atmospheric descent using trace gas measurements from remote sensors" for publication in ACP. The authors use the SD-WACCM model to argue that processes other than vertical advection of CO are important in the calculation of polar winter descent rates in the upper stratosphere and mesosphere. The paper is well written and the results will be of interest to the scientific community. I recommend publication after the following revisions.**

**—General comments— I echo here a comment made by another reviewer that the authors need to first show consistency between the CO measurements and the evolution of CO in SD-WACCM. If the CO tendencies between the obs and the model do not agree then it's not appropriate to use the obs to calculate w-corrected.**

Section 4 has been edited to include the calculations that were made with the modelled CO, instead of the measured CO. We agree that this is more consistent and avoids the differences between the model and the instruments (Table 2). The results led to the same conclusions because of the level of agreement between he modelled and measured CO, as stated in the original and edited manuscript.

**Overall, both in the abstract (maybe even the title), throughout the paper, and in the conclusions, the authors need to emphasize that these results are based on SDWACCM. Report quantitative error estimates to vertical motions derived from tracers instead of using provocative language like "found to be invalid".**

The referenced sentence, from the conclusion, has been edited to emphasise SD-WACCM and to clarify that the conclusion is indicated from the results:

*"An assessment using SD-WACCM indicates that a commonly used approximation of the vertical mean velocity of the atmosphere, $\overline{w*}$, using tracer (CO in this case) isolines is not valid, and an alternative interpretation of the rates derived from trace gas measurements is suggested: an effective rate of vertical transport for the given trace gas."*

The second line of the abstract states that SD-WACCM is what is used to assess the assumption of dominant vertical advection:

*"Using output from the Specified Dynamics Whole Atmosphere Community Climate Model (SD-WACCM) between 2008 and 2014, tendencies of carbon monoxide (CO) volume mixing ratios (VMRs) are used to assess a common assumption of dominant vertical advection of tracers during polar winter."*

The edit to use the SD-WACCM CO profiles instead of the measurements for calculating descent rates is now also mentioned in the abstract:
*"SD-WACCM CO profiles are combined with the CO tendencies to estimate errors involved in calculating the mean descent of the atmosphere from remote sensing measurements."*

The calculation of w*_corrected is somewhat crude (as mentioned in the paper and now expanded upon in the new manuscript), as it combines tendencies in the TEM formalism with values derived using CO VMRs in the atmosphere. These points are now emphasized in Section 4 and it is made clearer that the goal of w*_corrected is to get a qualitative estimate of the errors that may be incurred by assuming pure vertical advection when using tracers to calculate w*:
*"The resulting rate is called $w_{CO}$ corrected. This could be considered a crude approach, combining daily averaged CO output with CO tendencies calculated using the TEM formalism, but the aim here is to provide an estimate of the errors that may be incurred by neglecting influences on CO other than vertical advection. In any case, the results involving $w_{CO}$ corrected are discussed in a qualitative manner, instead of for quantitative error analysis."*

The conclusion contains the qualitative information from Section 4 about the estimated errors that are found using w_CO_corrected:
*"The differences between the two results are of the same order as the calculated rates, and the rates are prone to showing opposite directions for the mean vertical wind. The corrected rates more closely match the TEM vertical wind velocity from SD-WACCM, but both results using CO show smaller magnitudes relative to the TEM vertical wind, in agreement with the work of Hoffmann (2012b). The "true" rate of atmospheric descent appears to be masked by sinks of CO, and by transport processes that oppose the tendency due to vertical advection."*

A more quantitative analysis of the relative influence of tendencies is now included in Section 5 offering a clearer picture of the results as they relate to an approximation of pure vertical advection when deriving descent rates. The reader is also reminded of the daily tendencies showing significant variation on time scales of a week:
*"There are no months where the relative strength of other processes can be considered negligible compared to the relative strength of $adv\_w *$. The closest approximations of this situation are at 50 km altitude in October and at 46 km altitude in November, when other processes contributes 13.7 % and 9.6 % of $adv\_w *$, respectively. These percentages then vary significantly with altitude. For October, the value increases to 18.6 % at 46 km, 22.5 % at 60 km, and is 61.13 % at 80 km. For November, the value increases to 34.4 % at 54 km, and is 70.8 % at 80 km.*
*The results for the south polar average, in Fig. 6, are qualitatively similar to those for the north. The relative strength of $adv\_w *$ shows a maximum of ~0.8. Both hemispheres show a peak in chem at 80 km for most of winter (see Sect. 3.3). The relative strength of $Xedd$ is not as prominent at the south as the north, likely due to the higher stability of the southern polar vortex. The points at which the relative strength of other processes is smallest compared to $adv\_w *$ are at 56 km in April (8.3 %) and at 46 km in May (6.8 %). For April, the value increases*

*to 22.5 % at 46 km and 21.5 % at 66 km, and is 56.9 % at 80 km. In May, the value increases to 16 % at 54 km, and is 69.1 % at 80 km.*

*For the 10 days directly before and after SSWs, in Fig. 7, the relative strength of $adv\_w*$ is less than 0.5 at all altitudes. $Xedd$ is strong below 60 km, such that the relative strength of other processes has a larger magnitude than that of $adv\_w*$ at many altitudes. The relative strength of $adv\_w*$ shows a more oscillatory structure with altitude, and there is a local minimum at about 70 km in the data for 10 days after SSWs. There is also a positive peak in the relative strength of $Xk_{zz}$ after SSWs at this altitude.*

*Aside from considering what value would classify as negligible, the significant variation in strength of other processes compared to $adv\_w*$, over altitude, adds complexity to the method of following a tracer over an altitude range to determine the descent rate. One must also consider that while this section discusses monthly averaged data, tracers are often followed for several days to determine the changes in altitude over that time, and that the magnitudes of each tendency can vary significantly over this time scale (see Figures 3, 4, and 5)."*

**Figures 1, 3, and 4 are too small, bordering on illegible. In many cases all of the figure panels shown are neither introduced nor discussed. Please reduce the content of each of these figures to simply support the key point being made in the text.**

Figure 1 has been enlarged, changed to landscape layout, and edited to make the data clearer.

Figure 3 and 4 have been separated into three figures, according to the different scenarios assessed: above Kiruna, 80N zonal mean, and north polar average). The panels have also now been made larger.

For Figure 1: Section 2.4 indicates that Table 2 shows the correlation and regression coefficient for each of the panels in Figure 1. The caption for Figure 1 states that there are four altitudes shown (one for each panel), and indicates that more information can be found in Section 2.4 and in Table 2:

*Figure 1: Comparisons of daily CO VMRs from KIMRA, MLS, and SD-WACCM above Kiruna for 2008 through 2014. Values are displayed at 46, 56, 66, 76, and 86 km altitude. Correlation and regression coefficients for the datasets are given in Table 2. See Sect. 2.4 for details."*

For Figure 3: The caption incorrectly stated that the tendencies shown in the panels are described in Section 2.4. This should have said Section 3.1 and has been fixed:

*Figure 3: 11-day running mean tendencies of CO (in ppmv/day), calculated using daily averaged SD-WACCM output. Tendencies shown are for 67˚ N for the winters of 2008/2009 and 2010/2011. See section 3.1 for a description of the tendencies which are represented in the TEM continuity equation."*

Section 3.3 discusses each of the tendencies that are plotted in Figure 3. Each panel is not sequentially discussed, but rather the relevant points, as well as similarities and differences between the scenarios.

For Figure 4 (original manuscript): w_CO_corrected and the difference between that and w_CO were not mentioned in the caption. This has now been fixed:

*"Figure 6: Rates of vertical motion, in km/day, calculated by tracking CO VMRs over time. $w_{CO}$ is calculated using SD-WACCM CO profiles. $w_{CO}\ corrected$ is calculated using a combination of SD-WACCM CO profiles and TEM tendencies (see Sect. 4 for details). The difference between the two rates of descent is also shown. The results plotted are for above Kiruna for the winters of 2008/2009 and 2010/2011. Contour lines are spaced by 0.2 km/day. Areas with tightly packed contours (black areas) occur when there are very low CO VMRs and the calculation method is unreliable. White areas are where a CO VMR could not be tracked within the shown altitude range. The start date of the SSW on January $28^{th}$, 2009, is shown with a vertical green dashed line."*

**Figures 3, 4, and 8: swap the color bar to be blue for negative values and red for positives. Whenever possible, hold fixed the color bar range so that comparisons between winters and between tendency terms can be made.**
As the swapping of colour placement is a subjective matter, we will keep the current colour choice.

In originally making the figures, it was decided not to use the same colour range for each tendency because too much of the information is lost from the plots. Because of the varying values of the tendencies (within a winter and from year to year), there are times, for example, when all tendencies show relatively low values. With the same colour bar for each panel, the information about relative influence is then lost. There are too many instances of this nature to choose a single colour range that includes the relevant information. Instead, it was decided to use both labelled contours and colour bars. While it is admittedly not easy to quickly determine the relative magnitudes of the tendencies from a glance, relevant information is not omitted from the figure. The separation of the Figure 3 and Figure 4 into three, makes it easier for the reader to attain this information for each scenario in the edited manuscript.

**—Line-by-line comments—**

**Abstract Line 16 - "the relative importance of vertical advection is lessened: : :" – by how much? Give %**
The sentence has been edited to read
*"The relative importance of vertical advection is lessened, and exceeded by other processes, during periods directly before and after a sudden stratospheric warming, mainly due to an increase in eddy transport."*
The magnitudes of the processes are variable and to come up with a percentage would require choosing a single reference point with which to compare. The point being made here is that vertical advection cannot be considered as dominant, which is indicated by the fact that other processes become more important.

**1 Introduction Page 2, lines 25-35: ": : :defining the edges of the polar vortex is not straightforward." – cite Harvey et al. (2009) and Harvey et al. (2015)**
These references have been included in the edited manuscript.

**2.1 KIMRA, Page 3, line 29: does "average precision" mean daily average?**
Average precision refers to the fact that the precision varies somewhat from profile to profile. The sentence has been edited to say this and to give the approximate time resolution of a measurement:
*"The average precision (values can vary from one profile to another) of wintertime KIMRA CO VMRs range from 0.06 ppm at 46 km altitude to 2.7 ppm at 86 km. The average time resolution of a CO measurement is around 2 hours. KIMRA CO data presented in this work have been averaged to give daily profiles."*

**2.2. MLS, Page 4 line 7 - Does "highest pressure level" refer to the pressure level at the highest altitude?**
The sentence has been edited to clarify that is it is the pressure level at the highest altitude:
*"... and have a maximum (largest) precision of 11 ppm at the highest (in altitude) pressure level."*

**Line 8 - Does "averaged to produce daily profiles" in some spatial region?**
Yes, the information was originally in Section 2.4 but has been moved to Section 2.2:
*"MLS data presented here are within ± 2˚ latitude and ± 10˚ longitude of Kiruna, and have been averaged to produce daily profiles."*

**2.3 SD-WACCM – Given the fallibility in MERRA winds (mentioned in the intro) in the upper stratosphere and their inability to properly model the elevated stratopause in February of 2009, what (if any) impact does this have on SD-WACCM and the conclusions?**
Insofar as the actual winds are not well known in the upper stratosphere, then SD-WACCM may not reproduce the precise evolution of a given elevated stratopause event. However, WACCM can, and does, produce realistic elevated stratopause events (de la Torre at al., JGR, 2012; Chendran et al., JGAR, 2013), which is what counts as regards evaluating the effects of such variability on the behavior of CO.

The model can simulate accurately the statistical properties of elevated stratopause events, even if it does not simulate precisely the weather. The imprecise simulation of a specific elevated stratopause event does not materially change any of the conclusions of the paper.

In the comparison of CO from instruments and models, as in the manuscript, imprecise simulation of a specific elevated stratopause event may cause differences in the measured and modelled CO profiles. Looking at January 2009 in Figure 1 (enlarged version in the edited manuscript), SD-WACCM captures the quick decrease and subsequent increase in CO VMRs seen by both KIMRA and MLS at the time of the SSW. The differences seen between the CO profiles on a daily timescale are reflected in the regression coefficients and correlations listed in Table 2.

**2.4 CO VMR comparison, Page 5 Figure 1 is inadequate. Please compare the model and measured CO in a comprehensive way that convincingly demonstrates that CO**

**tendencies are in agreement.**
As the comment overlaps somewhat with the first general comment, there is some repetition in the answer.

Section 4 has been edited to include the calculations that were made with the modelled CO, instead of the measured CO. We agree that this is more consistent and avoids the differences between the model and the instruments (Table 2). The results led to the same conclusions because of the level of agreement between the modelled and measured CO, as stated in the original manuscript.

The aim of the paper is to use the individual tendencies of CO in the atmosphere to ascertain whether vertical advection can be considered dominant to such an extent that the atmospheric descent rates can be calculated by observing tracer motion. This cannot be achieved with only measurements, as there is not enough information to separate the contributions to the observed VMRs. A model that simulates accurately the observed evolution of atmospheric CO VMRs is used to separate the contributions, or tendencies.
The comparison of atmospheric CO measured by instruments and modelled with SD-WACCM is vital to determining whether the model accurately represents the CO VMRs that are observed in the atmosphere. Figure 1 has been enlarged, changed to landscape layout, and edited to make the data clearer. Table 2 lists the quantitative results of the comparison and shows the level of agreement between the model and the instruments, which is quite high for daily averages. A more in-depth comparison of KIMRA, MLS, and SD-WACCM has been made by Hoffmann et al. (2012), and the comparison is made in the current manuscript because there have been updates to the model and the datasets. Section 1 now includes this information:
*"The values are similar to those found for earlier versions of the model and data (Hoffmann et al., 2012a), with differences mainly due to updates to the modelled CO (Garcia et al., 2014) and the data products (Livesey et al., 2015; Ryan et al., 2017)."*

Section 1 has been edited to clarify the above points:
*"The aim of this study is to assess the limits of the above assumption when using tracer measurements from remote sounders to derive rates of vertical motion in the middle atmosphere. Measurements alone do not provide enough information to enable separation of the contributions to changes in tracer VMRs, and so an atmospheric model must be employed. The specified dynamics version of the Whole Atmosphere Community Climate Model (SD-WACCM) is used to determine the relative contributions to changes in CO VMRs during polar winter. The results are combined with daily average modelled CO to estimate the error associated with descent rates calculated assuming pure vertical advection of the tracer. Three commonly used representations of the data are assessed: a local area above a specific location (Kiruna, 67.8˚ N, 20.4˚ E, in this case), a zonal mean at a certain latitude (80˚ N is used as an example), and a polar mean (60˚ - 90˚ N). The winters of 2008/2009 and 2010/2011 are used in the study as an example of a winter with a strong SSW and a winter with a relatively stable vortex, respectively. The rate calculations were also performed using CO measurements from the Kiruna Microwave Radiometer (KIMRA) and the Microwave Limb Sounder (MLS) (not*

*shown), and the results lead to the same conclusion. This was expected due to the level of agreement found in a comparison of the modelled and measured CO (Sect. 2.4)."*

**Figure 2 – increase panel size and symbol size. Reword last line of the caption to be "Parcel positions on Jan 28th (start of the 2009 SSW) is indicated by black asterisks."**
The panels and symbols have been made larger and the sentence has been edited to above.

**3.3 Tendencies of CO during Arctic winter Page 6, line 28 – ": : :in depth analysis is not made as it is not the focus of the study." – Then can the results be summarized in fewer than 36 panels?**
Thorough analyses of the tendencies trace gases have been made as the sole focus of a paper (e.g., Monier and Weare, ACP, 2011). Such an analysis is not warranted in the current manuscript, but the tendencies are an essential part in understanding the processes that effect CO.

The figure has now been separated into three, according to the different scenarios. The different scenarios are used to reflect the predominant ways that the tracer data are used: a point measurement (from a ground-based instrument), a zonal mean at a specific latitude (most often from satellite data and sometimes from ground-based), and a north polar average or vortex average (from satellite data). The two winters are used as examples of a winter with a SSW and one without.
With the figures separate, the reader can now clearly see the tendencies for both winters of a scenario, and focus on one scenario if they so choose.

**Page 6, line 31 – ": : :decrease in CO VMRs" add "in the upper mesosphere" – Does this mean air is ascending there?**
The sentence has been edited to contain the addition.
While the vertical advection term refers to changes in CO due to ascent and descent of air, the molecular diffusion and eddy diffusion terms are diffusive in nature.
The effect of molecular diffusion on a trace gas can be regarded as the sum of two contributions: diffusion, which is proportional to the second derivative of the mixing ratio, and a vertical drift velocity, which is related to differences in the molecular mass of the trace gas compared to the molecular mass of the background atmosphere. The mass of CO is close to that of the background atmosphere and so CO has a small drift velocity.
The effect of eddy diffusion also depends on the gradient of the mixing ratio as well as the diffusion coefficient. The process is a related to the turbulent diffusion associated with gravity wave breaking.

**Page 7, first paragraph –**
**Mention different color scales. Give relative magnitudes wrt w\*, i.e., chem is 10% of w\*. Can we interpret negatives = ascent/poleward and positives = descent/equatorward (or is it not that simple)?**

The different colour scales are now mentioned, and there have been additions to the quantitative description of each tendency, and information added about what the sign of the advection tendencies indicates about the direction of air motion:

[revised manuscript text omitted]

**Page 7, first paragraph, line 7 – ": : :because of proximity to the edge of the polar vortex." – No, both are well inside the vortex core.**
The polar vortex is not a fixed structure in space; it moves and can be distorted.
Kiruna has been shown to be inside, outside, or in the edge of the polar vortex on many occasions (e.g. Kopp et al., JGR, 2003; Ryan et al., AMT, 2016, Ryan et al., ESSD, 2017).
The same has been shown for Eureka, which is at 80N (e.g., Bird et al., JGR, 1997; Batchelor et al., AMT, 2010).
One can expect Kiruna to experience more variability with respect to the edge of the polar vortex, due to planetary scale waves, for instance, because of its location in latitude.
Some of the references in Section 1 of the current manuscript, like from Manney et al., describe the movements and distortions/splitting of the north polar vortex.

**Page 7, line 14**
**– "There is also a brief change to a positive" – do you mean negative?**
Yes. Thank you. This has been fixed.

**Page 7, lines**
**19-20 – This is very useful. Please do this for all tendency terms.**
Quantitative information has been added for each tendency term, with the other additions to Section 3.3.

**4 Rates of vertical motion with KIMRA and MLS Page 7, lines 30-31 – ": : :the concentration is adjusted using the tendencies of the continuity equation: : :" add "from WACCM". This section will gain credibility after showing that the model and observed tendencies are in agreement.**
As the comment overlaps somewhat with an earlier comment regarding Section 2.4, there is some repetition in the response.

Section 4 has been edited to include the calculations that were made with the modelled CO, instead of the measured CO. We agree that this is more consistent and avoids the differences between the model and the instruments (Table 2). The results led to the same conclusions

because of the level of agreement between the modelled and measured CO, as stated in the original manuscript.

**Page 8, lines 17-20 – please reword. Are derived descent
rates stronger because they need to counteract the opposing terms?**
The lines have been reworded, and an additional sentence added for clarity:
*"… the values of $w_{CO}$ are generally of a smaller magnitude than $w_{CO}$ corrected during winter, meaning the calculated rates of descent are stronger if one accounts for CO tendencies other than vertical advection. This makes sense because, as seen in Figure 3, the other transport terms of the continuity equation (and the chemical loss term) tend to oppose the vertical advection term. In other words, the results indicate that the "true" rate of atmospheric descent is masked by sinks of CO, and by transport processes that oppose the tendency due to vertical advection."*

**Page 8,
line 23 – ": : :around the time of SSW is decreased: : :" could use an arrow in the figure
to highlight this location.**
A vertical dashed line has been added to mark the date. The caption has been edited to reflect this.

**Figure 4 – can any of this information be shown using a scatter plot with w\* along one
axis and w\*-corrected along another? The points could be colored by altitude. It is currently
very difficult to look at the panels and understand the comparisons quantitatively.
How often would you get points with opposite signs? Is it more likely to get different
directions up high or down low?**
As was stated earlier in relation to a previous comment, because the method combines daily averaged CO profiles and TEM tendencies, the aim of the calculations is to provide a qualitative error estimate. This is now emphasised in the text:
*"The resulting rate is called $w_{CO}$ corrected. This could be considered a crude approach, combining daily averaged CO output with CO tendencies calculated using the TEM formalism, but the aim here is to provide an estimate of the errors that may be incurred by neglecting influences on CO other than vertical advection. In any case, the results involving $w_{CO}$ corrected are discussed in a qualitative manner, instead of for quantitative error analysis."*

The text, which describes the results of Section 4, now explicitly states that three main qualitative points are made about the results.
*"There are three main qualitative points, common to each scenario and year, that are evident from the results.*
*First, the values of $w_{CO}$ are generally of a smaller magnitude than $w_{CO}$ corrected during winter, meaning the calculated rates of descent are stronger if one accounts for CO tendencies other than vertical advection. This makes sense because, as seen in Figure 3, the other transport terms of the continuity equation (and the chemical loss term) tend to oppose the vertical advection term. In other words, the results indicate that the "true" rate of atmospheric descent is masked by sinks of CO, and by transport processes that oppose the tendency due to vertical advection. Second, the differences between the two rates are often of the same order as $w_{CO}$.*

*Third, the signs of $w_{CO}$ and $w_{CO}$ corrected are often opposite, meaning the calculated direction of air motion is prone to change when accounting for CO tendencies other than vertical advection. In each example for 2008/2009, the magnitude of the positive (upward) motion around the time of SSW is decreased for $w_{CO}$ corrected compared to $w_{CO}$. After the SSW, and into March, the strongest descent values are seen around $70-80$ km in $w_{CO}$ corrected, compared to values of ascent seen in $w_{CO}$ at the same location."*

The panels are now larger because of the separation of the figure into the three scenarios., so the reader can better observe the times/altitudes when the two rates have different signs, which is indicated by them having different colours.

**Figure 5, 6, and Page 9 – refer to 60-90 as the "polar cap" – not the pole.**
"North pole" or "south pole" are not used in the edited manuscript. "… polar average" is used.

**6 Discussion, Page 10 line 2 - ": : :not a valid one." Add "according to SD-WACCM" line**
The line now reads:
*"As the results here using SD-WACCM indicate that the commonly used approximation of $\overline{w*}$ with $w_\chi$ (using tracer observations) is not valid, we suggest an alternative interpretation of $w_\chi$: as an effective rate of vertical transport for the trace gas $\chi$."*

**30 – Reword ": : :are representative for the complete mesospheric air: : :"**
The line now reads:
*"… but with an assumption that the overall dynamic effects on CO are representative for mesospheric air, and so $w_{CO}$ is representative of $w_\chi$ for all tracers."*

**7 Conclusion, Page 11, line 17 – ": : :no months during polar winter when vertical mean advection dominates the budget of CO to such an extent that vertical mean velocity can be accurately derived." – This statement is too strong. Instead, give error estimates as a function of altitude and time over the winter.**
The referenced sentence in the conclusion has been edited to emphasise that this conclusion is indicated by the results from all sections. Statements of the main results have been added/edited in the conclusion as follows:
*- "The results show that dynamical processes other than vertical advection cause non-negligible changes in CO VMRs during winter, and particularly directly before and after sudden stratospheric warmings."*
*- "Significant changes in CO tendencies from SD-WACCM occur on the order of days. The results also show a chemical sink for CO, present throughout polar night, due to the layer of night-time OH at approximately 80 km."*
*- "Rates of atmospheric motion were calculated when assuming only vertical advection, and corrected rates were calculated by including tendency information for all processes. The differences between the two results are of the same order as the calculated rates, and the rates are prone to showing opposite directions for the mean vertical wind."*
*- "The "true" rate of atmospheric descent appears to be masked by sinks of CO, and by transport processes that oppose the tendency due to vertical advection"*

- *"Monthly mean relative tendencies for CO show that the summed magnitude of processes other than vertical advection can constitute a large fraction of the changes in CO VMR. For a given month, the magnitude of the other processes, relative to vertical advection, changes by several tens of percent over the altitude range under investigation."*
- *"The results suggest that there are no months during polar winter when vertical mean advection dominates the budget of CO to such an extent that vertical mean velocity can be accurately derived within the altitude range."*

Section 5 now contains a quantitative description of the results and a discussion of the results that indicate the conclusion with respect to the monthly data, and also reference the significant changes in tendencies on time scales of several days that is seen in the earlier section:

*"There are no months where the relative strength of other processes can be considered negligible compared to the relative strength of $adv\_w *$. The closest approximations of this situation are at 50 km altitude in October and at 46 km altitude in November, when other processes contributes 13.7 % and 9.6 % of that of $adv\_w *$, respectively. These percentages then vary significantly with altitude. For October, the value increases to 18.6 % at 46 km, 22.5 % at 60 km, and is 61.13 % at 80 km. For November, the value increases to 34.4 % at 54 km, and is 70.8 % at 80 km.*

*The results for the south polar average, in Fig. 6, are qualitatively similar to those for the north. The relative strength of $adv\_w *$ shows a maximum of ~0.8. Both hemispheres show a peak in chem at 80 km for most of winter (see Sect. 3.3). The relative strength of $Xedd$ is not as prominent at the south as the north, likely due to the higher stability of the southern polar vortex. The points at which the relative strength of other processes is smallest compared to $adv\_w *$ are at 56 km in April (8.3 %) and at 46 km in May (6.8 %). For April, the value increases to 22.5 % at 46 km and 21.5 % at 66 km, and is 56.9 % at 80 km. In May, the value increases to 16 % at 54 km, and is 69.1 % at 80 km. For the 10 days directly before and after SSWs, in Fig. 7, the relative strength of $adv\_w *$ is less than 0.5 at all altitudes. $Xedd$ is strong below 60 km, such that the relative strength of other processes has a larger magnitude than that of $adv\_w *$ at many altitudes. The relative strength of $adv\_w *$ shows a more oscillatory structure with altitude, and there is a local minimum at about 70 km in the data for 10 days after SSWs. There is also a positive peak in the relative strength of $Xk_{zz}$ after SSWs at this altitude. Aside from considering what value would classify as negligible, the significant variation in strength of other processes compared to $adv\_w *$ over altitude adds complexity to the method of following a tracer over an altitude range to determine the descent rate. One must also consider that while this section discusses monthly averaged data, tracers are often followed for several days to determine the changes in altitude over that time, and that the magnitudes of each tendency can vary significantly over this time scale (see Figures 3, 4, and 5.)"*

---

## Author Response (AR2)

The authors express their gratitude to the referees for taking the time to offer useful criticism for the manuscript.

**The manuscript is much improved over the previous version, however I am still troubled by 2 issues.**

**First, the authors have, in the new Figures 6-8, replaced measured CO with SD-WACCM CO. The current figures are good, but it seems to me that it would very much strengthen the paper to show, in addition to w_CO from the SD-WACCM calculation, the w_CO values from the measurements that were shown in the previous version. In fact, this seems to me to be the only reason the measurements should be included in the paper. Judging from the figures that were presented in the previous version, w_CO (measurement) and w_CO (WACCM) look reasonably similar, and both quite dissimilar from the corrected w_CO. It seems to me that this is the main point of the paper.**

w_CO, calculated using KIMRA and MLS CO VMRs, is now included in Figures 6-8. The main text and captions have been edited accordingly. The following text has been added to Section 4:
*"$w_{CO}$ was also calculated using the data from KIMRA, for 67° N, and from MLS, for the zonal mean at 80° N and the north polar mean. The results are included in Figures 6 – 8 for comparison. The $w_{CO}$ values from SD-WACCM and from the instruments show good agreement, with some differences that would be expected due to the levels of agreement of the CO VMRs (see Sect. 2). Considering the calculations using SD-WACCM, there are three main qualitative points, common to each scenario and year, that are evident from the results. The same conclusions are reached when using KIMRA and MLS in place of SD-WACCM.*

This is not the only reason that measurements should be included in the paper. One must determine whether the model accurately represents the CO VMRs that are observed in the atmosphere. This has not previously been done for the current versions of the model and measurements, so a comparison was performed as part of this project and presented in Section 2.

The main point of the paper is expressed in the first three sentences of the abstract:
*"We investigate the reliability of using trace gas measurements from remote sensing instruments to infer polar atmospheric descent rates during winter within 46 – 86 km altitude. Using output from the Specified Dynamics Whole Atmosphere Community Climate Model (SD-WACCM) between 2008 and 2014, tendencies of carbon monoxide (CO) volume mixing ratios (VMRs) are used to assess a common assumption of dominant vertical advection of tracers during polar winter. The results show that dynamical processes other than vertical advection are not negligible, meaning that the transport rates derived from trace gas measurements do not represent the mean descent of the atmosphere."*

**Second, while I understand (although I disagree with) the authors wish to use different scales for the panels in Figures 3-5 for cases where there are very large differences between the different**

**mechanisms, the current complete reliance on self-scaling of the figures is simply sloppy. There is no reason why, for instance, the adv_w term shown for 2 years and 3 geographic regions should not be shown with the same scale in all 6 instances.**

5    The colourbar scale is now the same across Figures 3-5, while the labelled contours remain with values individual to each panel so that relevant information is retained when there are small values for each tendency, thus eliminating perceived simple sloppiness.

[revised manuscript text omitted]